# Eliminating blood oncogenic exosomes into the small intestine with aptamer-functionalized nanoparticles

Xiaodong Xie[1,5], Huifang Nie[1,5], Yu Zhou[1], Shu Lian[1], Hao Mei[1], Yusheng Lu[2], Haiyan Dong[1,3], Fengqiao Li[1], Tao Li[1], Bifei Li[1], Jie Wang[1], Min Lin[1], Chaihung Wang[1], Jingwei Shao[1], Yu Gao [1], Jianming Chen[2], Fangwei Xie[4] & Lee Jia[1,2]*

There are disease-causing biohazards in the blood that cannot be treated with modern medicines. Here we show that an intelligently designed safe biomaterial can precisely identify, tow and dump a targeted biohazard from the blood into the small intestine. Positively charged mesoporous silica nanoparticles (MSNs) functionalized with EGFR-targeting aptamers (MSN-AP) specifically recognize and bind blood-borne negatively charged oncogenic exosomes (A-Exo), and tow A-Exo across hepatobiliary layers and Oddi's sphincter into the small intestine. MSN-AP specifically distinguish and bind A-Exo from interfering exosomes in cell culture and rat and patient blood to form MSN-AP and A-Exo conjugates (MSN-Exo) that transverse hepatocytes, cholangiocytes, and endothelial monolayers via endocytosis and exocytosis mechanisms, although Kupffer cells have been shown to engulf some MSN-Exo. Blood MSN-AP significantly decreased circulating A-Exo levels, sequentially increased intestinal A-Exo and attenuated A-Exo-induced lung metastasis in mice. This study opens an innovative avenue to relocate blood-borne life-threatening biohazards to the intestine.

---

[1] Cancer Metastasis Alert and Prevention Center, College of Chemistry; Fujian Provincial Key Laboratory of Cancer Metastasis Chemoprevention and Chemotherapy, Fuzhou University, Minjiang University, Fuzhou, Fujian 350116, China. [2] Institute of Oceanography, Minjiang University, Fuzhou, Fujian 350108, China. [3] Fujian Key Laboratory for Translational Research in Cancer and Neurodegenerative Diseases, Institute for Translational Medicine, School of Basic Medical Sciences, Fujian Medical University, Fuzhou, Fujian 350108, China. [4] Department of Oncology, Fuzhou General Hospital, Fuzhou, Fujian 350001, China. [5] These authors contributed equally: Xiaodong Xie, Huifang Nie. *email: cmapcjia1234@163.com

**B**lood substances are usually eliminated from the systemic circulation through two routes: renal filtration with excretion into the urine and hepatobiliary clearance into the small intestine through Oddi's sphincter[1]. Substances <6 nm are usually eliminated by renal filtration through glomeruli[2]. Larger substances that are not immediately trapped by liver Kupffer cells are usually eliminated via hepatobiliary clearance into the small intestine. Blood flows through the hepatic artery and portal vein with oxygen and nutrients into the liver and flows out through the hepatic vein and inferior vena cava, leaving some circulating substances behind that can be translocated through the fenestrated vascular endothelium into the Disse space to be transcytosed through hepatocytes. Finally, these substances converge from bile canaliculi into the bile duct for excretion[3]. These eliminated substances include drugs of MW 290–1300 Da, IgA and insulin-like growth factor as we previously reported[1]. We wondered if we could eliminate these disease-causing biohazards (viruses, inflammatory factors, fat and others) from the blood into the small intestine. Thus many diseases with unmet medicinal agents can be treated in this way, or the root causes of diseases may be eradicated.

Artificially intelligent nanomaterials can identify their targets (enzymes, receptors, cells and others) in vivo and selectively bind them[4]. We demonstrated that dual antibodies or dual aptamers conjugated together can capture the rare circulating tumour cells (CTCs) in vivo with high specificity due to their ability to identify two biomarkers on the CTCs and seize them with double hands[5–7]. Drugs can bind circulating human serum albumin, and the binding changes their elimination rate[8]. Based on these findings, we hypothesise that an appropriate safe biomaterial may be functionalized to specifically bind an unwanted biohazard in the blood, and extract it into the small intestine via the sphincter of Oddi. The size and biomembrane properties of the circulating oncogenic exosomes make them a perfect model material to test this hypothesis.

Exosomes are extracellular vesicles generated and released by all cells. Nano-sized extracellular vesicles (40–150 nm) are naturally present in the blood, and participate in cell-to-cell communication. The size and their membrane lipid bilayer facilitate their efficient entrance into other cells[9,10]. Oncogenic exosomes colonise in specific organ sites and initiate pre-metastatic niche formation through their integrin-mediated fusion with organ-specific resident cells[11,12]. The surface of exosomes is rich in various surface biomarkers, including tetraspanins (CD9, CD63, CD81 and CD82) and the epidermal growth factor receptor (EGFR; or ErbB-1; HER1 in humans) from the originating cell[13]. EGFR is an important transmembrane protein activated by the binding of its extracellular domains to its specific ligands[14,15]. Mesoporous silica nanoparticles (MSNs) are a relatively safe biomaterial that can be rapidly uptaken by the liver, and quickly excreted from the liver into the gastrointestinal (GI) tract[16,17]. MSNs meet the least prerequisites for a candidate material to function as the tractor to tow the bound exosomes out of the circulation system into the GI tract. In this study, we chemically functionalized MSNs with EGFR-targeting aptamers (MSN-AP) for identifying and binding the exosomes derived from human lung cancer cells A549 (A-Exo) that express high levels of EGFR[18,19]. The intermolecular binding forces, including recognition and charge forces, between MSN-AP and A-Exo make the hepatobiliary elimination of A-Exo possible. The creative concept and the biotechnology approaches presented here open an avenue to dump the unwanted circulating biohazards into the small intestine.

## Results

**Exosome engineering and characterisation.** We isolated A-Exo from the supernatant of A549 cell cultures by ultracentrifugation.

To analyse the dynamic biodistribution of exosomes, we transfected A-Exo with a foreign DNA sequence (Fig. 1a) using an exosome transfection kit to quantify the intracellular and tissue distribution of the transfected A-Exo at low concentrations by using quantitative RT-PCR. To examine whether the designed MSN-AP could specifically distinguish the target A-Exo from the non-target exosomes, we isolated and fabricated control exosomes from human embryonic lung fibroblast culture medium (HELF; designated as H-Exo) using the same protocol.

Exosome contents, sizes and surface biomarkers vary depending on their cell origins[13]. We used high-resolution atomic force microscopy imaging (AFM; Fig. 1b, c) and dynamic light scattering to determine the average size of the exosomes. We used transmission electron microscopy (TEM; Fig. 1d, e) to characterise their external appearance. The average diameters of H-Exo and A-Exo were 97.9 nm and 130.1 nm, respectively, with good homogeneity (Fig. 1g). The exosomes appeared round- or oval-shaped (Fig. 1d, e) with zeta potentials −32.4 mV (A-Exo) and −33.8 mV (H-Exo) (Supplementary Fig. 1a, b). The mean protein concentrations of H-Exo and A-Exo were 58.1 μg per ml and 46.1 μg per ml, respectively, back-calculated from the absorbance-protein concentration standard curve (Fig. 1h). The nanoparticle tracking analysis (NTA) results proved that each microgram of exosomes contains $2 \times 10^7$ exosome particles. The amount of foreign DNA in exosomes after transfection was 9.4 fmoles (A-Exo) and 39.4 fmoles (H-Exo) per microgram of exosome protein analysed by combining the BCA protein assay with a quantitative RT-PCR method (Fig. 1h; Supplementary Fig. 1c). Exosomes were characterised using a flow cytometry backgating strategy based on CD9, CD63 and EGFR biomarkers after the exosomes bound to their corresponding aldehyde sulfate beads or secondary antibodies. The flow cytometer equipped with a cell sorter showed that CD9 and CD63 expression in A-Exo was 28.4 and 8.3%, respectively, versus 22.2 and 19.5% in H-Exo. Interestingly, only A-Exo expressed a high level of EGFR (50.5%), while H-Exo expressed no significant level of EGFR (0.8%; Fig. 1f).

**Functionalization and characterisation of MSNs.** To investigate the effects of the MSNs charge on its tissue distribution and cell trafficking, we synthesised the MSNs with both a positive and negative surface charge by reacting hexadecyl trimethyl ammonium chloride (CTAC) with triethanolamine (TEA) to form surfactant templates in ethanol as described previously[20,21]. 3-Aminopropyltriethoxysilane (APTES) and tetraethyl orthosilicate (TEOS) were added to form amination product MSN-NH₂, which was then reacted with succinic anhydride to form carboxyl-modified MSN-COOH. To make the functionalized MSN recognise and bind the targeted exosomes, we conjugated the anti-EGFR DNA aptamer (AP) to both the positively charged MSN (MSN-AP) and the negatively charged MSN (MSN-AP−) in the presence of 1-ethyl-3-(3-dimethyl-aminopropyl) carbodiimide (EDC) and N-hydroxysuccinimide (NHS) after labelling the AP with the fluorophore Cy3 for MSN-COOH; (MSN-AP-Cy−) or Cy5 for MSN-NH₂ (MSN-AP-Cy) for the visualisation of MSN-AP. The MSNs were then functionalized with amino groups and reacted with the carboxyl groups of AP using the cross-linking reagents NHS and EDC (Fig. 2a). The partial fluorophore and AP on the MSNs can be protected by the' highly ordered hexagonal pore structure and adjustable pore size (1.5–10 nm) of the MSNs[22].

TEM imaging showed a uniform hexagonal shape of the modified MSNs with average diameters of ~70 nm (Fig. 2b–d), which was further confirmed by AFM measurements (Fig. 2e–g) showing no significant difference in size distributions between

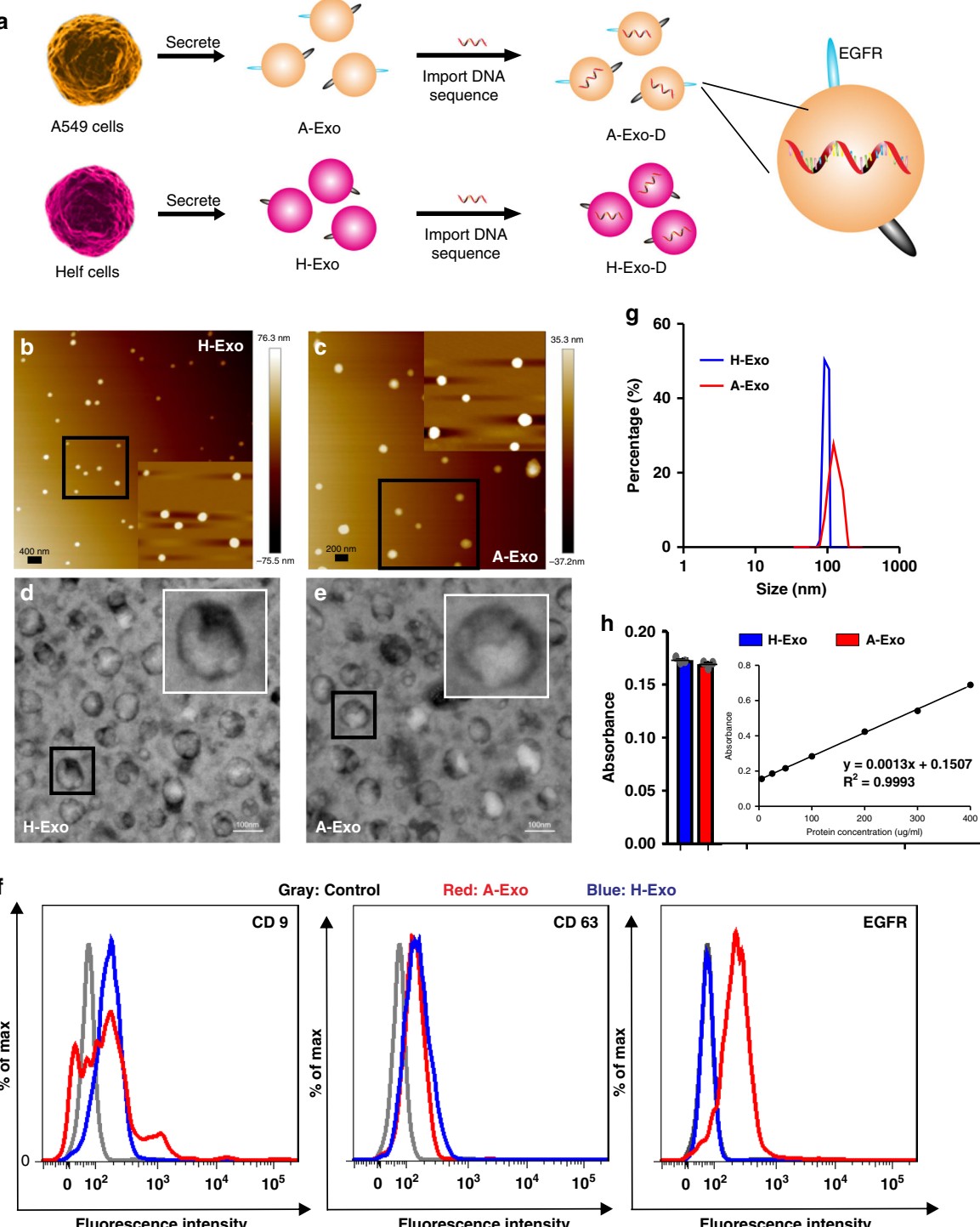

**Fig. 1** Engineering and characterising extraneous exosomes. **a** Exosomes derived from either the human lung cancer cell line A549 (A-Exo expressing high EGFR) or the control human lung fibroblast cell line HELF (H-Exo) were isolated and transfected with a DNA sequence for quantitative analysis. **b**, **c** High-resolution atomic force microscopy (AFM) and **d**, **e** transmission electron microscopy (TEM) show the size and morphology of A-Exo and H-Exo, respectively, with an additional sixfold amplification in size (insets). **f** Flow cytometry analysis showing the expression levels of CD9 and CD63 on A-Exo and H-Exo with high EGFR expression on A-Exo only. **g** Size distribution and **h** protein quantification using the standard concentration curves of A-Exo and H-Exo ($n = 3$ independent samples). Scale bars: 400 nm (**b**), 200 nm (**c**), 100 nm (**d**, **e**). Data presented as the mean ± s.e.m. Source data are provided as a Source Data file.

MSN, MSN-AP− and MSN-AP (Fig. 2h). We used the NTA method and determined that each milligram of MSNs contained $7 \times 10^{10}$ MSN nanoparticles. However, the zeta potentials for MSN-COOH, MSN-AP−, MSN-NH$_2$ and MSN-AP were −34.6, −35.8, +16.7 and +15.4 mV, respectively, at the physiological pH

of 7.4 (Supplementary Fig. 2c–f). Carboxylation and amination changed the potential levels of the MSNs significantly, but conjugation of the negatively charged DNA aptamers to the MSNs shifted the potential slightly. The number of aptamers on the MSNs were 0.041 and 0.043 nmol per mg for MSN-AP-Cy3

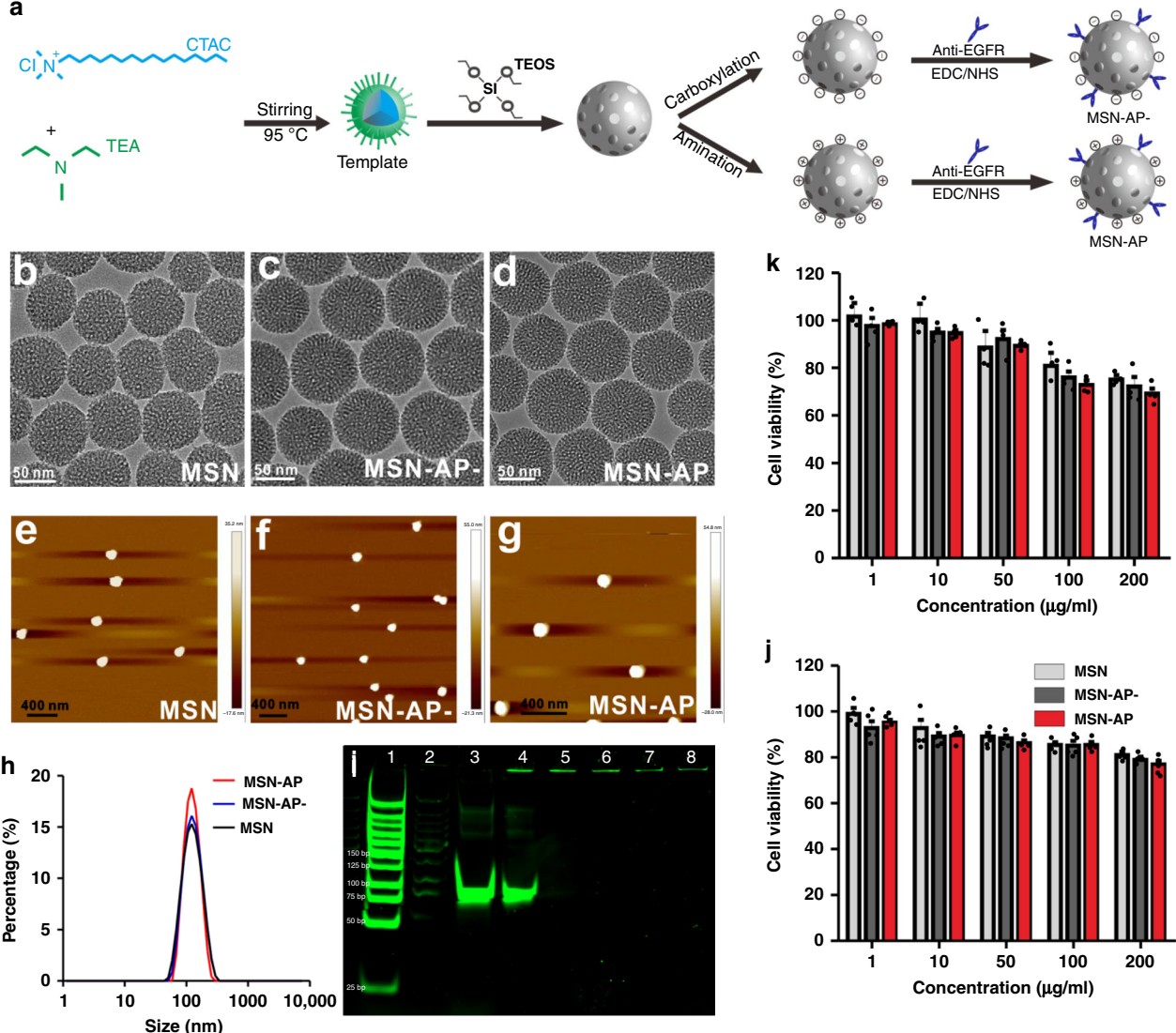

**Fig. 2** MSN functionalization and characterisation. **a** Schematic showing the synthesis and surface modification of MSN with EGFR-targeting aptamers following carboxylation (negative charge; MSN-AP−) and amination (positive charge; MSN-AP). **b–g** TEM (**b–d**) and AFM (**e–g**) images of MSN, MSN-AP− and MSN-AP. **h** Size distribution. **i** PAGE electrophoresis showing 1, markers; 2, MSN; 3, free aptamer; 4, MSN mixed with AP; 5, MSN-AP-; 6, MSN-AP; 7, MSN-AP- after 8-h incubation in blood; 8, MSN-AP after 8-h incubation in blood. **j** Effects of MSN, MSN-AP− and MSN-AP on viability of endothelial cells (**j**) and human hepatocyte LO2 cells (**k**). n = 5 reduplications. Data presented as the mean ± s.e.m. Source data are provided as a Source Data file.

and MSN-AP-Cy5, respectively, as measured by the fluorescent calibration curves of Cy3 or Cy5 (Supplementary Fig. 2a, b). Based on the information, we determined that each single MSN particle contains 344 aptamers. The MSN-AP conjugates were stable in whole blood for at least 8 h (37 °C) as measured by polyacrylamide gel electrophoresis (PAGE; Fig. 2i) as we previously performed[7]. The FTIR data demonstrated the existence of aptamers on the MSN (Supplementary Fig. 3).

The cytotoxicities of MSN, MSN-AP− and MSN-AP to both human umbilical vein endothelial cells (HUVECs; Fig. 2j) and human normal hepatocytes (LO2; Fig. 2k) were measured at concentrations up to 200 μg per ml. Although MSN, MSN-AP− and MSN-AP showed a concentration-related decrease in cell viability with increased downregulation of LO2 (Fig. 2j, k), the biomaterials seem relatively safe as we and others demonstrated previously in in vitro and in vivo settings[20,22].

**In vitro binding between MSN-AP and A-Exo.** We then investigated whether MSNs labelled with the red fluorophore Cy3

or Cy5 could specifically recognise and bind exosomes labelled with PKH67 (green fluorophore). Incubation of MSN-AP− or MSN-AP with A-Exo and H-Exo, respectively, in cell medium (Fig. 3a–d) showed specific binding between MSN-AP and the exosomes by both confocal microscopy and flow cytometry analyses. The merged yellow dots show the binding and attraction between the MSNs and exosomes after their co-incubation in cell medium (Fig. 3a, b, far right). The positively charged MSN-AP bound more A-Exo more did MSN-AP− (Fig. 3a, b; far right and lower panels). Cationized MSN without aptamers lacks the ability to specifically recognise and bind to negatively charged exosomes (Supplementary Fig. 4). The binding was further verified by flow cytometry. Both MSN-AP and MSN-AP− specifically bound to A-Exo with little binding to H-Exo, which expresses low EGFR. The results also indicate that the recognition and binding of positive MSN-AP to A-Exo is approximately threefold stronger than that of negative MSN-AP− to A-Exo (Fig. 3d, 26.6% versus Fig. 3c, 8.1% positive events). To further test the binding between MSN-AP and A-Exo, and the resulting bound conjugate MSN-

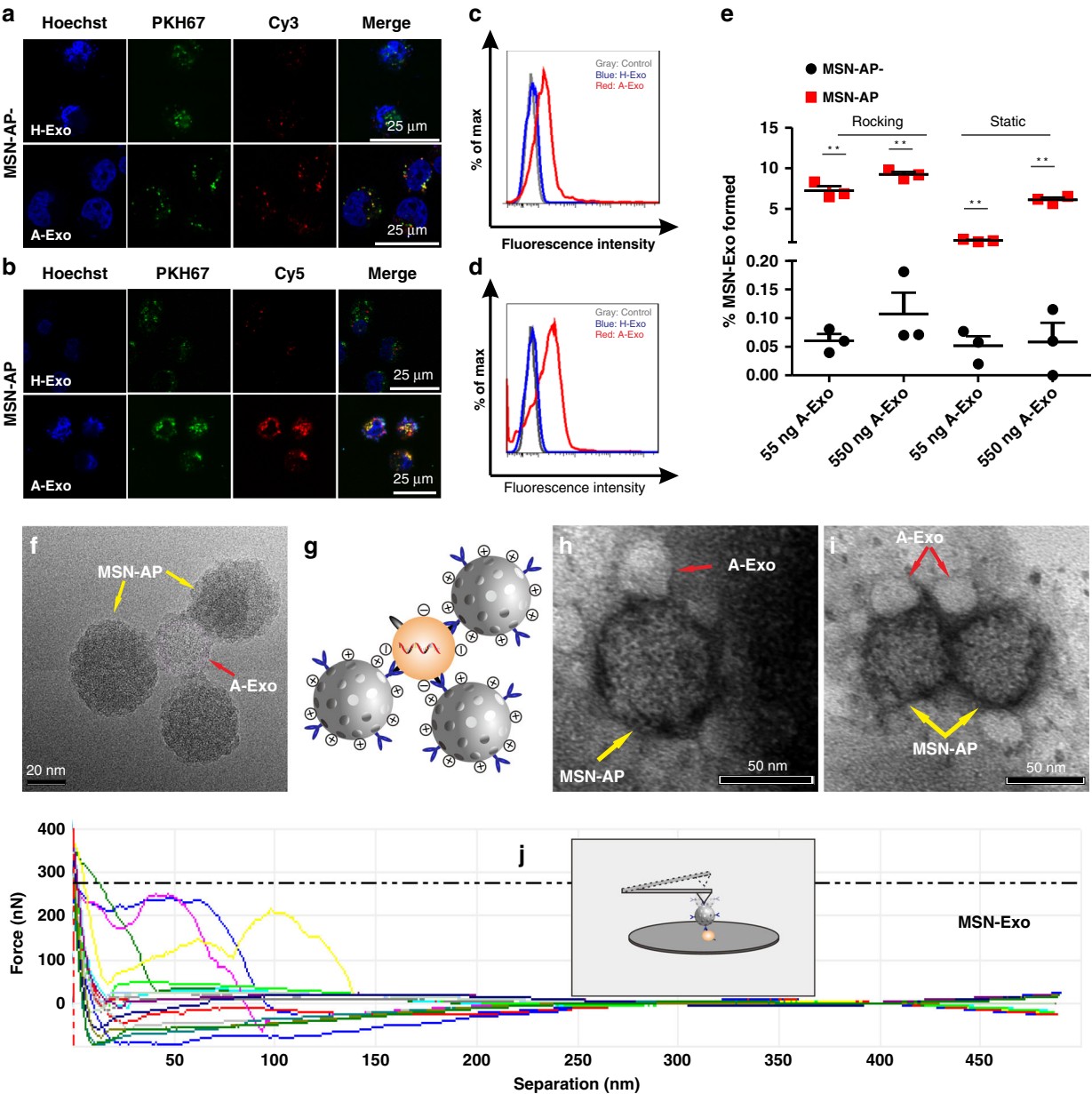

**Fig. 3** Recognition and binding between MSN-AP and A-Exo in cell media and rat blood. **a**, **b** Fluorescence microscopy of the binding between A-Exo (labelled with green PKH67) and MSN-AP− (**a**; labelled with red Cy3) or MSN-AP (**b**; labelled with red Cy5) using H-Exo as a control (upper panel). Note the significant binding between A-Exo and MSN-AP (far right and lower panel in **b**). **c**, **d** Flow cytometry analysis showing binding between A-Exo and MSN-AP− (**c**), or MSN-AP (**d**) using H-Exo as a control. **e** Flow cytometry quantification showing the percentage of A-Exo captured by MSN-AP- or MSN-AP to form MSN-Exo in rat blood after 1 h of static or rocking incubation at 37 °C. There was no significant binding between MSN-AP and H-Exo, which lacks EGFR. $n = 3$ independent samples. **f** TEM image showing the binding between MSN-AP and A-Exo in cell medium. **g** Schematic showing the molecular binding between the EGFR aptamer on MSN-AP and the EGFR receptor on A-Exo, as well as the electronic attraction between the positively charged MSN-AP and negatively charged A-Exo. **h–i** TEM images showing conjugation between MSN-AP and A-Exo in blood without (**h**) and with (**i**) rocking. **j** Single-molecular force spectroscopy of AFM shows the intermolecular force of the MSN-Exo conjugate corresponding to the separation distance; the inset depicts that the AFM cantilever tip attaches to MSN-Exo. Data presented as the mean ± s.e.m. **$P < 0.01$, unpaired two-tailed $t$ test. Source data are provided as a Source Data file.

Exo, we incubated MSN-AP with A-Exo in rat blood at 37 °C under both static and rocking conditions, and quantified the intermolecular binding between MSN-AP and A-Exo in 300 µl of blood by flow cytometry. At the fixed concentrations of MSN-AP − and MSN-AP (both 50 µl, 100 µg per ml), the number of bound MSN-Exo particles increased when more A-Exo was incubated with MSN-AP− and MSN-AP. The binding product MSN-Exo was obvious under both static and rocking conditions (100 rpm) with more MSN-Exo produced from MSN-AP than from MSN-AP−

incubation with MSN-AP− (Fig. 3e, Supplementary Fig. 5), indicating that the positively charged MSN-AP attracts and binds A-Exo more favourably than the negatively charged MSN-AP− does.

We then analysed the intermolecular recognition and binding forces between MSN-AP and A-Exo. The basic molecular mechanics include covalent bonds and noncovalent bonds. The latter describe long-range electrostatic and van der Waals forces, and account for electronic polarizability. We used the most

simplistic formula, i.e., Hooke's law[23] $v(l) = k(l-l0)2/2$, to evaluate intermolecular forces, where $k$ is the force constant (the stronger the bond, the higher the value of the force constant), $l$ is the intermolecular distance at equilibrium and $l0$ is the distance when all other terms are set to zero. The inset of Fig. 3j depicts an AFM cantilever to which MSN-AP is attached and bound to A-Exo on a silicon wafer. The force profile reflects the increasing force applied to the MSN-Exo bond when the intermolecular distance reaches a maximum until the bond suddenly breaks. The average intermolecular force was 200 nN at a maximum distance of ~70 nm, representing both the recognition force and charge force of the MSN-Exo bond. TEM images further show binding between MSN-AP and A-Exo when upon incubation at 37 °C in cell media (Fig. 3f) and blood (Fig. 3h) or in blood after 4 h of rocking at 100 rpm (Fig. 3i). The TEM images were taken after centrifugation and washing of the samples (Fig. 3h, i). Ultra-high-resolution field-emission scanning electron microscopy (SEM) images (Supplementary Fig. 6) with a wide-field view were also used to observe the binding. This result indicates the important role of electrostatic attraction in forming MSN-Exo (Fig. 3g).

To further prove the formation of the MSN-Exo in blood and its endocytosis by hepatocytes, we incubated A-Exo (labelled with or without PKH67) with Cy5-labelled MSN-AP-labelled in rat blood with rocking at 100 rpm for 4 h at 37 °C, and collected the supernatant after centrifugation (200 g, 5 min). We then added the supernatant (containing the conjugated MSN-Exo, free MSN-AP-Cy and A-Exo labelled with PKH67) to confocal dishes covered with LO2 hepatocytes. The supernatant (containing A-Exo, MSN-Exo and MSN-AP-Cy) was also spun at 15,000 g for 10 min, and the precipitate was incubated with CD9-coated beads for flow cytometry analysis of the MSN-Exo formed in rat blood after two washes of the MSN-Exo-conjugated CD9 beads with PBS buffer (Fig. 4a). Figure 4b shows the intact MSN-Exo in yellow endocytosed by LO2 cells. MSN-AP could not recognise and bind to the normal exosomes in the rat blood (Supplementary Fig. 7).

**MSN-Exo conjugate traverses liver cells.** The liver is mainly composed of two types of epithelial cells: hepatocytes that are polarised and account for 60–80% of parenchymal cells and cholangiocytes, which are epithelial cells that line intrahepatic bile ducts and account for 3–5% of the liver cell population. Additional cells include Kupffer cells that make up 80–90% of the total body macrophage population[24,25].

To investigate whether the conjugate MSN-Exo could traverse these cells, we incubated PKH67-labelled A-Exo with MSN-AP-Cy− or MSN-AP-Cy at 37 °C for 1 h and then added the incubation solution to LO2 human hepatocytes grown on confocal microscopic dishes. We used time-lapse image sequences to track the trafficking of the formed MSN-Exo or PKH67-labelled A-Exo alone in LO2 cells for up to 600 min. Figure 4d–e clearly show the trafficking and exocytosis of A-Exo (PKH67 green), MSN-AP-Cy− (Cy3 red), MSN-AP-Cy (Cy5 red) and MSN-Exo (yellow) by LO2 cells. LO2 cells took up exosomes easily (Supplementary Fig. 8).

The endocytosed MSN-Exo in yellow moved as a single entity within the LO2 cells. MSN-Exo was finally excreted from the cell ~20 min later (Fig. 4d, e; Supplementary Fig. 9b, c, lower panels). We also tested the permeability of MSN-Exo in other cells (Kupffer cells, endothelial cells and intrahepatic cholangiocytes) under the same conditions (Supplementary Figs. 10–15). To further demonstrate that the endocytosed MSN-Exo can be excreted by LO2 as a single entity, we centrifuged (15,000 g) the above incubation mixture that contained MSN-Exo, A-Exo and MSN-AP, and collected the precipitate, which was the LO2 cells

containing MSN-Exo and/or MSN-AP only. We incubated the LO2 cells on the transwell membrane, and collected the cell medium from the lower chamber of the transwell. Confocal microscopy analysis (Fig. 4c) showed the merged yellow dots that were the MSN-Exo excreted from the LO2 cells as a single entity. Red dots (representing MSN-AP-Cy) were also found (Supplementary Fig. 9a).

To analyse the dynamic uptake/transcytosis of the MSN-Exo through liver cells, we incubated LO2 hepatocytes, endothelial cells and cholangiocytes (epithelial cells that line the intrahepatic bile ducts) on the transwell. The pre-incubated mixture of MSN-AP-Cy− or MSN-AP-Cy with A-Exo, in which MSN-Exo was formed mainly by +/− charge attraction and had reached the association–dissociation equilibrium, was added to the apical side of the cells on the transwell membrane, and the permeated solution from the lower chambers of the transwell was collected for permeability analysis[26]. To determine the effect of liver Kupffer cells on the uptake/transcytosis of MSN-Exo through liver cells, we co-incubated Kupffer cells with LO2 or endothelial cells at a cell number ratio of 1:6 before the addition of the pre-incubated mixture of MSN-AP, MSN-AP− and A-Exo to the individual co-incubation system (Fig. 4f). We measured the number of MSN-Exo conjugates formed in each sample in the lower chambers of the transwell at the assigned time points. Figure 4g–k and Supplementary Figs. 16–20 show the dynamic transcytosis of MSN-Exo across these liver cell lines quantified by the related fluorescence intensity of the fluorophores. MSN-Exo, MSN-AP and A-Exo permeated cholangiocytes, endothelial cells and LO2 cells easily (Fig. 4i–k). However, in the presence of Kupffer cells[27], the number of MSN-Exo conjugates across the cell monolayers was significantly reduced (Fig. 4g, h). Further confocal microscopy time-lapse shooting analysis revealed that these nanomaterials were engulfed and digested by Kupffer cells. Supplementary Figs. 10 and 11 show the dynamic degradation of MSN-Exo within Kupffer cells after 600 min of incubation. By analysing the numbers of transcytosed nanomaterials and their half-life in the presence and absence of Kupffer cells, we determined that >95% of the nanomaterials were engulfed by Kupffer/LO2 or Kupffer/endothelial co-incubated monolayers, and the rest could traverse across the sinusoidal plasma membrane and the canalicular membrane into the bile duct. However, the Kupffer/LO2 and Kupffer/endothelial co-incubation models may not truly reflect in vivo models.

To further explore the mechanisms by which MSN-Exo is endocytosed and exocytosed as a single entity by LO2 cells (Fig. 4l), we incubated LO2 cells with PKH26-labelled MSN-Exo in the presence and absence of the endocytosis inhibitor dynasore (50 μM, 1 h) or exocytosis inhibitor Bafilomycin A1 (100 nM, 6 h). Dynasore is a cell-permeable and commonly used selective endocytosis inhibitor that works via inhibition of dynamin GTPase activity[28]. Bafilomycin A1 is an exocytosis inhibitor for phosphonate-modified MSNs[29]. Dynasore treatment significantly reduced intracellular MSN-Exo by fourfold compared with PBS treatment, as evidenced by both flow cytometry (Fig. 4m, n) and confocal microscopy (Supplementary Fig. 21a). Bafilomycin A1 treatment significantly enhanced intracellular MSN-Exo by 1.2-fold, and decreased the exocytosed MSN-Exo compared with PBS treatment, as shown by flow cytometry (Fig. 4o, p) and confocal microscopy (Supplementary Fig. 21b, 22). These results indicate that MSN-Exo can transcytose through LO2 cells, and endocytosis and exocytosis are a main channel for MSN-Exo trafficking across LO2 monolayers.

**In vivo kinetic distribution and excretion of Exo.** To determine the in vivo excretion of the circulating A-Exo to the small intestine, we injected A-Exo or H-Exo (10 μg) into the rat portal

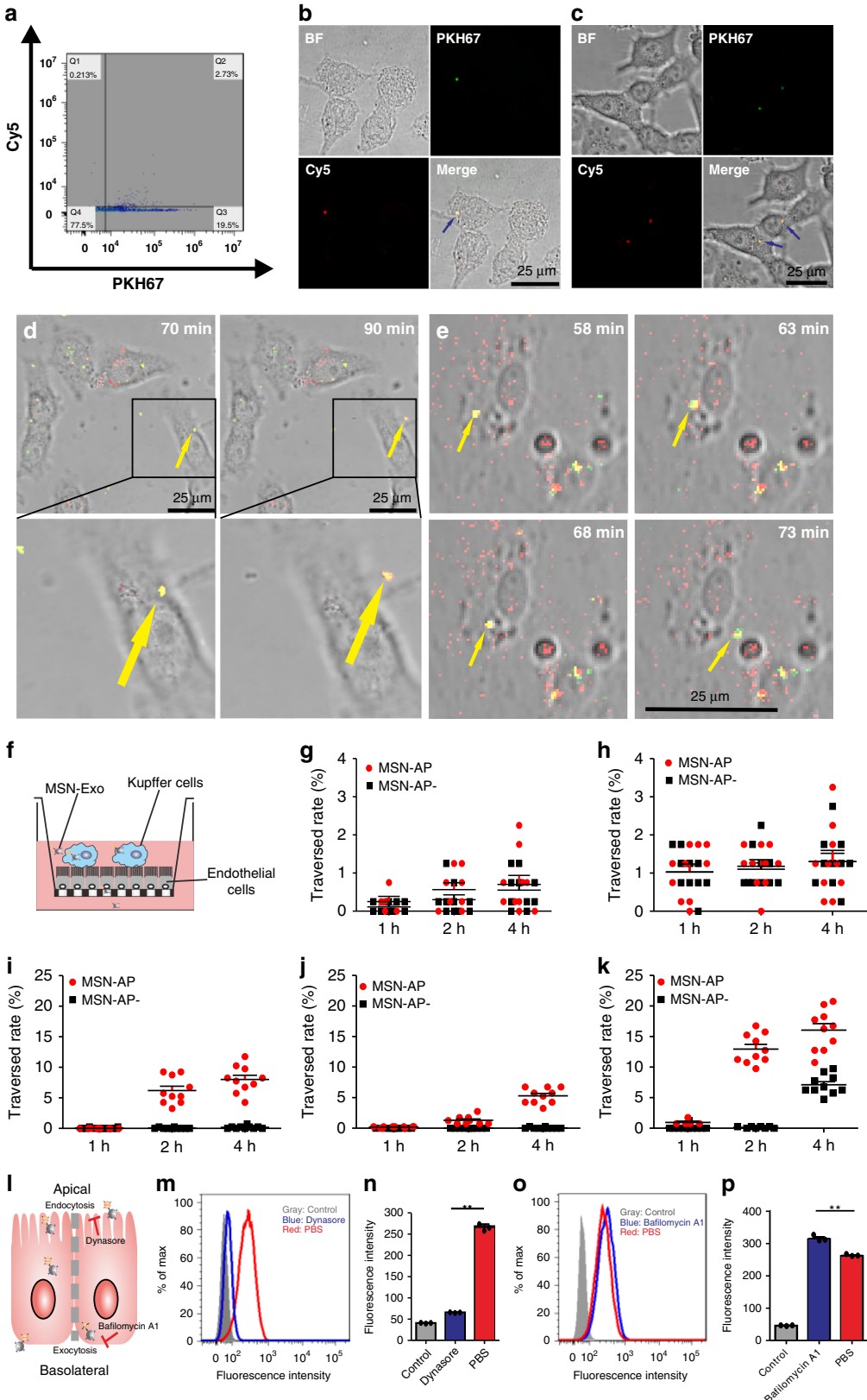

vein followed by the local injection of MSN-AP (5 mg per kg) to bind A-Exo and collected the duodenum contents from the rats 10 min later. We used the qRT-PCR method to quantify the DNA-transfected A-Exo content (Fig. 1a; Supplementary Fig. 23) in the duodenum after the injection, and found that MSN-AP significantly enhanced the amount of A-Exo accumulated in the

rat duodenum (Fig. 5a). However, MSN-AP in the portal vein failed to tow H-Exo into the duodenum (Fig. 5a, left 2 bars), indicating that the molecular recognition and binding between MSN-AP and exosomes can be specific and are important for excretion of the targeted biomaterial. Further systemic experiments were performed to determine whether MSN-AP could

**Fig. 4** In vitro conjugation between MSN-AP and A-Exo and their dynamic trafficking through liver cells. **a** Flow cytometry analysis showing MSN-Exo formed after rocking incubation of Cy-5-labelled MSN-AP with PKH67-labelled A-Exo in rat blood (37 °C, 4 h). **b** Confocal microscopy showing MSN-Exo formed (arrow) inside LO2 hepatocytes after incubation of red Cy-5-labelled MSN-AP with green PKH67-labelled A-Exo in rat blood. **c** The biostability of the conjugated MSN-Exo (arrows) after 4 h of incubation inside LO2 cells on a transwell. **d** Less MSN-Exo formed after 1 h of incubation of negative MSN-AP− with A-Exo. The confocal microscopy time-lapse image sequences show the trafficking of the MSN-Exo within the same LO2 cell. **e** More MSN-Exo formed after a 1 h incubation of positive MSN-AP with A-Exo. Note that the endocytosis and transcytosis of the same MSN-Exo occurred within the same LO2 cell recorded by the sequential time-lapse images. Yellow dots are the formed MSN-Exo; red dots, MSN-AP or MSN-AP−; green dots are A-Exo or H-Exo. **f** Transwell model that simulates the hepatobiliary biolayers, where the traversed molecules (MSN-Exo, MSN-Exo−) are collected from the transwell lower chamber for analysis. **g** Kupffer/LO2 cells co-incubated. **h** Kupffer/endothelial cells co-incubated. **i** Hepatic cholangiocyte monolayer. **j** Endothelial cell monolayer. **k** LO2 monolayer. Note the differences in $Y$-axis scale between **g**, **h** and **i**–**l** a model depicts transcytosis of MSN-Exo across hepatocytes. Data presented as the mean ± s.e.m. ($n = 10$). **m**–**p** Flow cytometry analyses show that the endocytosis inhibitor Dynasore decreases endocytosed MSN-Exo (**m**, **n**), while the exocytosis inhibitor BafilomycinA1 increases intracellular MSN-Exo (**o**, **p**, $n = 3$ independent experiments). Data presented as the mean ± s.e.m. **P < 0.01; one-way ANOVA (**n**, **p**). Source data are provided as a Source Data file.

reduce blood A-Exo. We injected 50 μg of DNA-transfected A-Exo into rat blood through the tail vein followed by injections of 200 μl of MSN-AP (5 mg per kg) or an equal volume of physiological saline. We collected bile from the common bile duct from 0–100, 100–200, 200–300 and 300–400 min to quantify the amount of A-Exo in the bile using qRT-PCR. Figure 5b shows that A-Exo in the bile increased over time. MSN-AP can significantly accelerate the excretion of the circulating A-Exo into the intestine.

After this success, we injected 10 μg of A-Exo and H-Exo labelled with PKH67 into mice through the tail vein. Ten minutes later, we injected saline, MSN, MSN-AP− or MSN-AP (5 mg per kg) into the mice. At 10 min, 2 h and 4 h after the injections, the mice were anaesthetised, and blood was collected for fluorescence microscopy quantification of blood exosomes. In general, blood MSN and MSN-AP levels reached their peaks immediately after i.v. administration, and declined quickly to ~5% of the original amount by 1 h after injection as we had observed previously regarding blood MSN dynamics[21]. It has been reported that 10 min after a bolus injection, most of the MSN-NH$_2$ analogues were distributed in the liver and then migrated into the duodenum 60 min later[16]. Figure 5c shows the blood levels of A-Exo and H-Exo following injections of saline, MSN, MSN-AP− and MSN-AP. These extraneous exosomes in the blood declined, and redistributed to the tissues. Interestingly, blood MSN-AP and MSN-AP− significantly accelerated the elimination of A-Exo, but not H-Exo, probably due to the specific recognition and binding between A-Exo and MSN-AP. However, there was no significant difference between MSN-AP− and MSN-AP in capturing blood A-Exo in vivo (Fig. 5c). This insignificance may be caused by the strong blood shear stress and the in vivo complexity that together interfere the difference between MSN-AP− or MSN-AP capture of A-Exo in vivo. Interestingly, when a large number of MSNs were distributed and concentrated in the liver (Fig. 5d; 5e far right), where the shear stress was low, and the nanomaterial velocity was reduced 1000-fold[25,30], this led to more interactions and binding between MSN-AP and A-Exo or H-Exo, and a significant difference in binding between A-Exo and MSN-AP in the livers was observed (Fig. 5e far right). MSN-AP did not recognise and bind to the other blood exosomes (Supplementary Fig. 24). Immunofluorescent co-staining demonstrated that macrophages and epithelial cells in tissues such as the lungs and livers took up PKH26-labelled exosomes (Supplementary Fig. 25). These lung cancer-derived exosomes were distributed to the lungs, spleens and livers at the levels much higher than those to the hearts and kidneys (Fig. 5d, e; note the different $Y$-axis scales), suggesting organotropism of A-Exo to these organs[31]. The blood MSN-AP helped A-Exo, but not H-Exo, distribute to the livers, and then redistribute to the small intestine as shown by the sequential changes in A-Exo levels in the organs, in particular,

the livers and small intestines, at 2 and 4 h in a dynamic manner as evidenced by the in vivo immunofluorescent co-localisation staining (Fig. 5d, e; Supplementary Figs. 26–30).

Figure 6a explains how MSN-AP recognises and binds the targeted blood exosomes together and transverses through the liver microenvironment into the bile duct and small intestine via the sphincter of Oddi. The surface positive MSN-AP could utilise its advantage of electrostatic attraction to bind the negative surface of A-Exo and hepatocytes to complete the process. We integrated the dynamic changes in blood A-Exo with changes in the liver and small intestine, and found that the decrease in blood A-Exo in the presence of MSN-AP was sequentially accompanied by a decrease in the liver A-Exo followed by an increase in A-Exo in the small intestine (Fig. 6b). The integrated data support our hypothesis that blood MSN-AP could conjugate the targeted A-Exo and the two compounds could transverse together via systemic circulation into hepatobiliary excretion.

We then examined whether the decrease in blood A-Exo could attenuate A-Exo-induced lung cancer metastasis. We subcutaneously implanted A549 cells into nude mice for 2 weeks until subcutaneous neoplasia occurred. The mice were intravenously administered saline, MSN, MSN-AP (both 5 mg per kg) and A-Exo (2 μg), every 3 days (starting on day 14 after the A549 implantation) for additional 3 weeks. Figure 6c–e shows the oncogenic exosome-induced organotropic metastasis of A549 CTCs to the mouse lungs (other organs are shown in Supplementary Fig. 31). Three week continuous injections of A-Exo prepared pre-metastatic niches to facilitate lung-specific metastasis of A549 CTCs in the mice, which could be inhibited by MSN-AP treatment.

To further test specificity of the MSN-AP (labelled with Cy-5) to bind human cancer exosomes, we added MSN-AP (final concentration of 40 μg per ml) to blood samples obtained from eight lung cancer patients after approval by the hospital ethics committee and measured the formation of MSN-AP-bound exosomes by using flow cytometry after the sample incubation/centrifugation process. The MSN-AP-bound EGFR-exosomes were identified and quantified in patient blood via flow cytometry gating based on the Cy5 fluorescent signals (Fig. 6f; Supplementary Fig. 32). Each sample was divided into 3–5 parts for repeated measurements (Fig. 6g). The number of EGFR-positive exosomes seemed to be well correlated with patient cancer stages. For example, late-stage patient 8 showed a significantly higher level of exosomes in compared with the patients at stage II.

## Discussion

Our circulation system is often contaminated with many unhealthy biomaterials including viruses such as HIV, CTCs, oncogenic exosomes and others, which cannot be controlled by

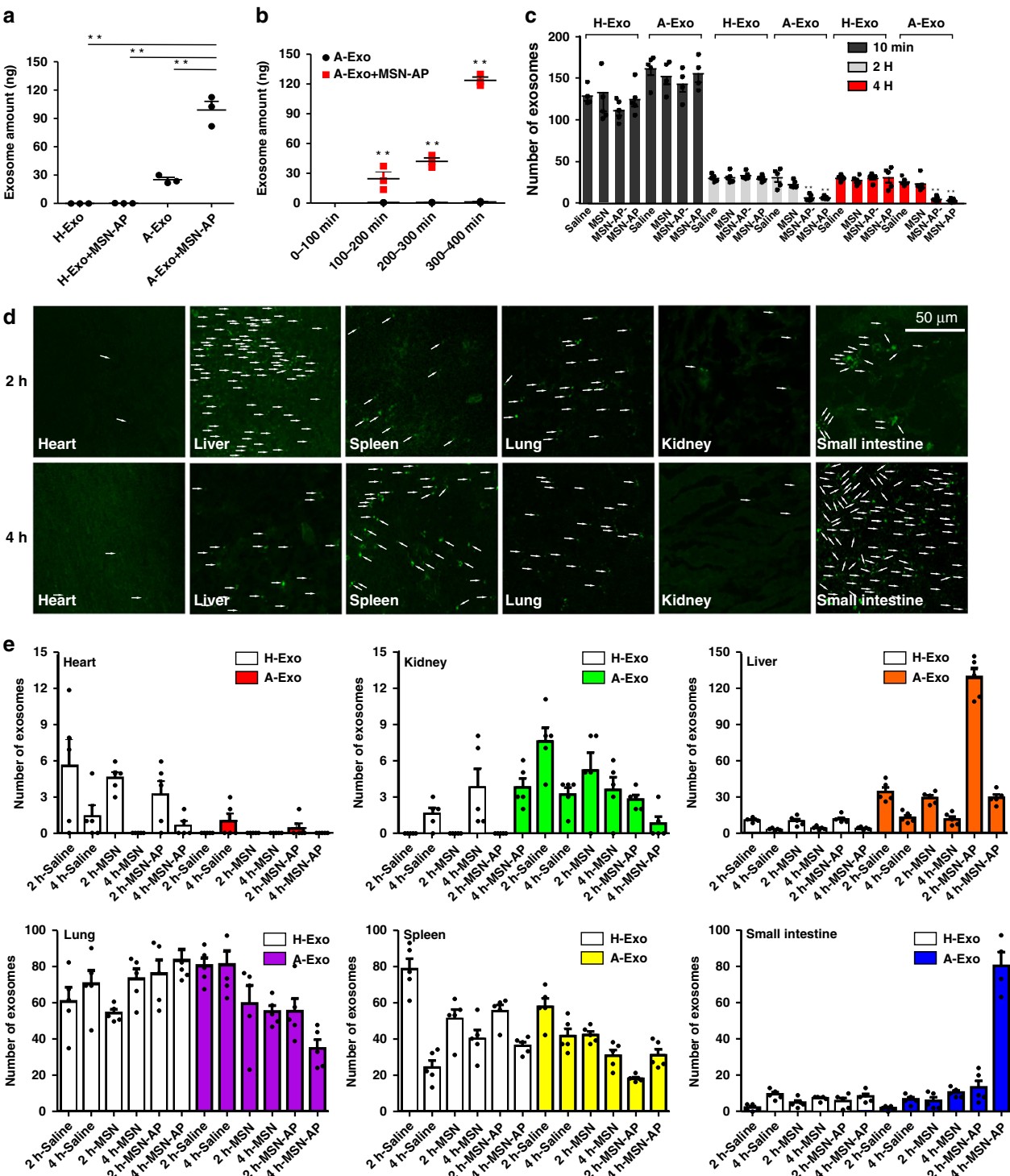

**Fig. 5** In vivo conjugation between MSN-AP and A-Exo accelerates hepatobiliary excretion of circulating A-Exo. **a** Excreted amounts of A-Exo or H-Exo in the small intestines after injection (both 50 μg) into the portal vein of rats (local injections) with or without MSN-AP injections (5 mg/kg, n = 3 rats). **b** A-Exo amount in rat bile after injection into the tail vein of rats (system injection) with or without MSN-AP injections. n = 3 rats. **c** Blood amounts of H-Exo or A-Exo (green PKH67-labelled) after tail vein injection into mice followed by injections of saline, MSN, MSN-AP− and MSN-AP (200 μl of 5 mg per kg), respectively. n = 5. **d** Confocal microscopy showing the PHK67-labelled A-Exo distribution (arrows indicate exosome foci) in various mouse tissues followed by MSN-AP injection (see Supplementary Figs. 26 and 27 for MSN, MSN-AP−, MSN-AP treatments). **e** Quantification of i.v. A-Exo distribution in tissues following saline, MSN, MSN-AP−, and MSN-AP i.v. administration compared with the H-Exo distribution. Note that the decrease in the A-Exo amount in the liver was followed by its sequential increase in the small intestine. n = 5. Data presented as the mean ± s.e.m. **P < 0.01; one-way ANOVA (**a**) and unpaired two-tailed t test (**b**). The Y-axis of Fig. 5c, e represents the average number of 4–5 randomly selected single fields of vision. Source data are provided as a Source Data file.

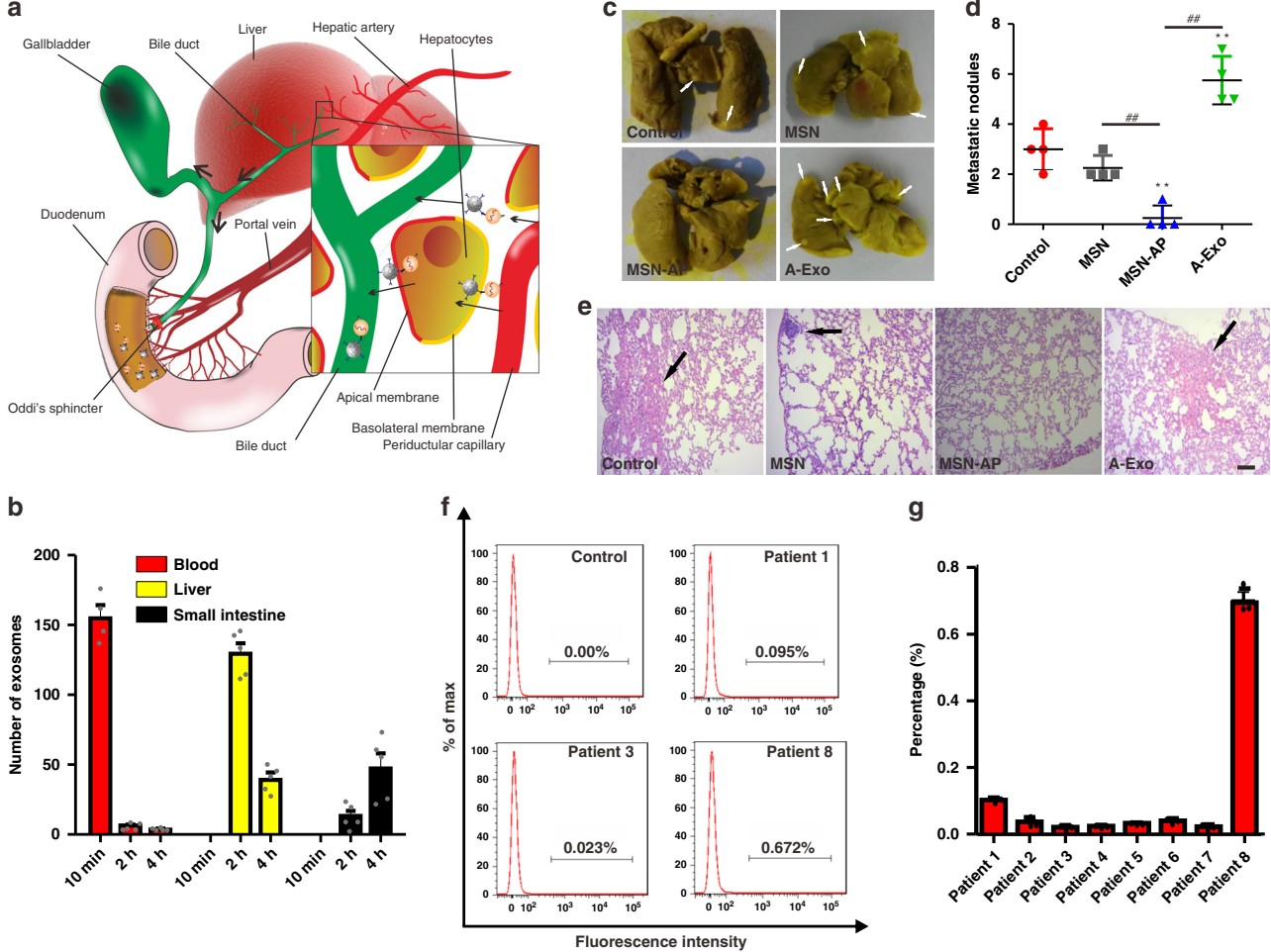

**Fig. 6** Elimination or deactivation of circulating exosomes by MSN-AP in animals and patient blood. **a** Schematic showing that MSN-AP binds to and tows circulating exosomes in the liver into the space of Disse, and the conjugated MSN-Exo can be endocytosed by polarised hepatocytes, transcytosed through the hepatocytes and enter the bile duct and small intestines via the sphincter of Oddi. **b** Dynamic decrease in blood A-Exo was sequentially accompanied by an increase of A-Exo in the small intestines of mice when MSN-AP was intravenously administered. $n = 5$. **c** Photos and **d** quantification of lung metastatic nodules developed (arrows) following subcutaneous implantation of A549 cells in nude mice receiving intravenous A-Exo (2 µg), saline, MSN or MSN-AP (both 5 mg per kg) every 3 days starting on day 14 after A549 implantation for an additional 3 weeks. $n = 4$ mice. **e** Lung H&E stains to show tumours. Scale bars: 200 µm. **f** Flow cytometry analysis and **g** quantification of patient EGFR-exosomes captured by MSN-AP. Note that patient 8 was in a late stage of lung cancer. $n = 8$ patients. Data presented as the mean ± s.e.m. **, ##, $P < 0.01$; one-way ANOVA (**d**). The $Y$-axis of Fig. 6b represents the average number of five randomly selected single fields of vision. Source data are provided as a Source Data file.

drugs, antibodies and vaccines because of severe side effects and resistance. This study exemplifies a safe means to eliminate these unhealthy biomaterials from systemic circulation. To eliminate an endogenous biohazard out of body, three prerequisites must be met:

First, the towing material must be safe, and itself can be eliminated. MSN meets this prerequisite. It is relatively safe[32]. When intravenously administered, 18% of the circulating MSN immediately distributed to the liver, causing a fast reduction in blood MSNs levels[21], and 60 min later, a significant percentage of MSN was eliminated into the duodenum via the biliary duct[16,17]. Most MSN reached the liver partially by a liver sequestering mechanism[25].

Second, the towing material can specifically identify and tightly bind the targeted biohazard. Previously, we developed dual-antibody or dual-aptamer conjugates that showed enhanced specificity for identifying circulating CTCs with an enhanced ability to tightly bind CTCs[5,7]. In this study, we demonstrate that the towing material MSN-AP can distinguish oncogenic exosomes from the H-Exo interfering exosomes (Fig. 3) or from the

blood of lung patients (Fig. 6f, g), and bind the oncogenic exosomes tightly even under shaking conditions (Fig. 3e). Figure 5 further demonstrates the in vivo binding between MSN-AP and A-Exo. The binding force may come from the electrostatic attraction between the positively charged MSN (+15 mV) and the negatively charged exosomes (−33 mV)[33], and the binding between EGFR aptamer on the MSN and the EGFR receptor on the oncogenic exosomes. There are multiple adhesive proteins on the surface of exosomes that may produce additional binding force between MSN-AP and A-Exo[34].

Third, the bound conjugate can pass through the hepatobiliary layers and enter the bile duct. We demonstrated that the bound conjugate MSN-Exo could transverse the polarised hepatocytes intracellularly (Fig. 4d, e). MSN-Exo could also transverse hepatocytes, cholangiocytes and endothelial monolayers in the presence of Kupffer cells although Kupffer cells, engulf some MSN-Exo (Fig. 4f–k). The in vivo data further demonstrated that MSN-AP significantly increased the biodistribution of oncogenic exosomes to the liver and sequentially to the small intestine (Figs. 5e, 6b). In addition, the fusion ability and organotropism of

the conjugated exosomes may contribute to the hepatobiliary traversing of MSN-Exo[31].

Overall, based on our previously developed biotechnology for in vivo capturing the rare CTCs, this study further demonstrates that the intelligently designed safe biomaterials can precisely identify, bind and tow the targeted biohazard from the blood into the bile duct for elimination. This study opens a horizon to explore and ultimately define a general mechanism for dumping disease-causing biohazards in the blood into the small intestine via Oddi's sphincter as long as the molecular structure and physiochemical properties of the biohazard are known.

## Methods

**Cell culture**. Human lung carcinoma A549 cells, HELF cells and Kupffer cells were cultured in RPMI 1640. Human hepatocyte (LO2) cells and hepatic cholangiocyte (HIBEpiC) cells were cultured in DMEM containing high glucose. All media contained 10% FBS and 1% penicillin–streptomycin. The cells tested negative for mycoplasma. A549, HELF and LO2 cells were purchased from the Cell Bank of Type Culture Collection of Chinese Academy of Sciences (Shanghai, China). HIBEpiC cells were obtained from Shanghai Bioleaf Biotech Co., Ltd. (Shanghai, China). Kupffer cells were prepared from cell suspension (100 g, 4 °C for 5 min) after collagenase perfusion of SD rat livers[35]. The supernatant was subjected to a 45% Percoll (Sigma-Aldrich) density-gradient centrifugation to remove the other cells, and the retained cells were resuspended. The supernatant was further centrifuged at 100 g, 4 °C for 5 min. The resulting supernatant was centrifuged at 350 g and 4 °C for 10 min to remove the fraction. The cells were suspended in Williams' Medium E and layered on a density cushion of 25/50% Percoll gradient and centrifuged at 900 g, 4 °C for 20 min. The cells in the middle of the two layers were collected and washed twice with Williams' Medium E. All animal operations were carried out under diethyl ether anaesthesia to avoid animal suffering as much as possible.

HUVECs were prepared after digesting the human umbilical vein with collagenase for 15 min, and maintained in 1% gelatin-coated tissue culture flasks in M199 (Gibco) medium supplemented with 20% FBS, 8 units per ml heparin, 100 mg per ml ECGS and 1% penicillin–streptomycin[36].

**Exosome isolation**. The isolation of exosomes from the cell culture medium was performed as described previously[31]. Briefly, cells were grown in medium containing 10% exosome-depleted FBS (Gibco, Thermo Fisher Scientific) and 1% penicillin–streptomycin. Exosomes were removed from 3 to 4 days cell culture supernatant by centrifugation at 500 g for 10 min. To remove any possible cell debris, the supernatant was spun at 12,000 g for 20 min. The supernatant was then ultracentrifuged at 120,000 g at 4 °C for 1 h. The exosomes were washed with PBS and ultracentrifuged at 120,000 g at 4 °C for another 1 h. The purified exosomes were then analysed and used for all experiments. We also used exosome preparation kits (System Biosciences) for exosome isolation.

**Exosome labelling**. To quantify exosomes, we fluorescently labelled the exosomes with PKH67($\lambda_{ex}$ = 490 nm and $\lambda_{em}$ = 502 nm, Sigma-Aldrich) or PKH26 ($\lambda_{ex}$ = 551 nm and $\lambda_{em}$ = 567 nm, Sigma-Aldrich). For exosome labelling, we mixed the exosomes with Diluent C regent (Sigma-Aldrich; 20 µl each). One microlitre of PKH67 was mixed with another 40 µl of Diluent C reagent. The above two solutions were mixed and incubated for 5 min, followed by the addition of an equal volume of 1% BSA to stop the staining. The labelled exosomes were washed with PBS, and collected by ultracentrifugation. Exosomes were also transfected with a foreign DNA sequence (5′-gtt ggc tgg tgc tgt taa tac cct cat gcg gtt aga aga tgg acg cct gc-3′) as a marker for in vivo quantification. The transfection was conducted using an exosome transfection kit (Weihui Bio, Beijing). First, 25 µl of TransExo buffer and 5 µl of TransExo reagent were mixed. Then, 10 µM foreign DNA sequence in 25 µl of TransExo buffer was added into the mixture, which reacted for 5 min at room temperature. Second, 50 µl of 40 µg per ml exosomes was added to the solutionfollowed by incubation overnight at 37 °C. Finally, the untransfected foreign DNA sequence was eliminated by the addition of 10 µl of DNase I (1 mg per ml) for 10 min. The removal efficiency of the untransfected foreign DNA sequence was determined using qRT-PCR. The product was DNA-transfected exosomes named A-Exo (from A549 cells) or H-Exo (from HELF cells).

**Exosome characterisation**. The size distribution of the exosomes was determined by a Malvern Instruments Zetasizer HS III (Malvern, UK). The size and surface appearance of exosomes were measured by AFM and TEM. The protein concentrations of the exosomes were measured by BCA protein assay (Pierce, Thermo Fisher Scientific). The number of exosomes per mg was determined by NTA. A flow cytometer with sorting function (Becton Dickinson FACS AriaIII cell sorter) was used to determine the expression levels of CD9, CD63 and EGFR on the exosomes extracted from HELF and A549 cells. In brief, 30 µl (40 µg per ml) of each H-Exo and A-Exo were mixed with 10 µl of Exosome-Human CD9 Flow

Detection beads (Invitrogen, Life Technologies), and rocked for 15-min at room temperature. The mixture was then diluted to 1 ml with PBS and stirred for 30 min at room temperature. The reaction was terminated with the addition of 100 mM glycine and 2% BSA in PBS and stirring for another 30 min. The mixture was washed twice with PBS, centrifuged at 15,000 g for 5 min and blocked with 10% BSA. After washing, the exosome-bound beads were incubated with 3 µl of anti-CD9 antibody (Abcam, EPR2949, ab92726), anti-CD63 antibody (Abcam, C-terminal, ab230414), anti-EGFR antibody (Abcam, EP38Y, ab52894), at 4 °C for 1 h. Exosome-bound beads were centrifuged at 15,000 g for 5 min and washed twice with PBS. The secondary antibody (Abcam, Goat anti-rabbit IgG H&L (FITC) ab6717) at a 1:500 dilution was used for 30 min at 4 °C. Secondary antibody incubation alone was used as the control.

**qRT-PCR-based analysis of exosomes**. The foreign DNA sequence transfected into exosomes (Fig. 1a) was determined by using qRT-PCR. The amplicon was generated by using the following primers: forward 5′-GTT GGC TGG TGC TGT TAA-3′ and reverse 5′-GCA GGC GTC CAT CTT CTA-3′. A series of concentrations ($10^{-2}$, 10, $10^4$, $10^5$ and $10^6$ fM) of the foreign DNA was used to establish the standard curve for DNA quantification of the exosomes.

**Transfected DNA extraction for exosome quantification**. The foreign DNA transfected into the exosomes was extracted from the tested biological samples using a Genomic DNA Extraction Kit (Takara, Japan). Briefly, 50 µl (40 µg per ml) of exosomes were added into a 1.5-ml tube, and then 180 µl of buffer GB, 20 µl of Proteinase K and 10 µl of RNAse (10 mg per ml) were added to the tube, and the mixture was incubated at 56 °C for 10 min. Then, 200 µl of 100% ethanol was added to the resulting mixture and evenly blended. A spin column was installed on the collection tube, and the mixture was moved to the spin column for centrifugation at 12,000 rpm for 2 min. The supernatant was discarded after centrifugation. Buffer WA (500 µl) and Buffer WB (700 µl) were added into the spin column followed by centrifugation at 12,000 rpm for 1 min. The supernatant was again abandoned. Then, the spin column was installed on a new 1.5-ml tube, and 100 µl of elution buffer was added to the centre of the spin column membrane. After standing for 5 min, the tube was centrifuged at 12,000 rpm for 2 min. The DNA sample was collected from the bottom of the tube.

**MSN functionalization**. The amino- and carboxyl-modified MSNs were synthesised as described previously[21,37]. Briefly, 2 g of hexadecyl trimethyl ammonium chloride (CTAC) and 0.1 g of triethanolamine (TEA) were dissolved in 20 ml of distilled water at 95 °C under rigorous stirring for 1 h. After stirring, 1.5 ml of TEOS was added to the mixture, which was stirred for another 1 h. The products were centrifuged at 15,000 g for 15 min and washed three times with ethanol to remove the residual reactants. The resulting products were dispersed in 50 ml of ethanol, and 200 µl of APTES was added to the solution. The resulting solution was refluxed for 4 h. After centrifugation and washing with water, the amino-functionalized MSNs (MSN-NH₂) were re-dispersed in ethanol and reacted with succinic anhydride for 12 h. The carboxyl-modified MSN (MSN-COOH) was collected and washed with water.

**Preparation of the aptamer-modified MSN**. Anti-EGFR DNA aptamers were conjugated to the surface of MSNs. In short, 2.8 nmol of amino-functionalized or carboxyl-functionalized DNA aptamers (5′-TAC CAG TGC GAT GCT CAG TGC CGT TTC TTC TCT TTC GCT TTT TTT GCT TTT GAG CAT GCT GAC GCA TTC GGT TGA C-3′)[38] were conjugated to 20 mg MSN-COOH or MSN-NH₂, respectively, by adding 15 mg of EDC and 15 mg of NHS, and stirring at room temperature for 24 h. The products were centrifuged at 15,000 g for 15 min and washed twice with distilled water to remove the unbound aptamers and the excess EDC and NHS. The final products, namely the aptamer-modified MSN (negative charge MSN-AP− and positive charge MSN-AP), were collected by centrifugation at 15,000 g at 4 °C for 10 min and washed twice with distilled water to remove the unbounded aptamers and excessive EDC and NHS. Cy3-labelled MSN-AP− (MSN-AP-Cy−) and Cy5-labelled MSN-AP (MSN-AP-Cy) were similarly synthesised by conjugating Cy3- or Cy5-labelled aptamers to the surface of the MSN. The amount of aptamers conjugated to MSNs was determined by measuring the fluorescence intensity of the supernatant (Cy3 at $\lambda_{ex}$ 548 nm and $\lambda_{em}$ 562 nm; Cy5 at $\lambda_{ex}$ 648 nm and $\lambda_{em}$ 668 nm) and back-calculated from an established standard curve. The number of MSN-AP particles per mg was determined by NTA. The conjugation between MSN and the aptamer was confirmed by Fourier Transform Infra-Red (FTIR) spectroscopy.

The zeta potentials and size distributions of MSN, MSN-AP− and MSN-AP were measured by a Malvern Zetasizer, and their physicochemical properties were characterised by TEM (200 kV) and AFM.

**In vitro cell viability assay**. Effects of MSN, MSN-AP−, and MSN-AP on the viability of the human normal cells were evaluated by the standard MTT assay. HUVECs and LO2 cells were cultured in 96-well plates at a density of 8000 cells per well. After 24 h of culture, the cells were incubated with a series of concentrations (1, 10, 50, 100 and 200 µg per ml) of MSN, MSN-AP−, MSN-AP at 37 °C for another 24 h. The cells were washed with PBS (pH 7.4) and incubated in culture

medium containing MTT (0.5 mg per ml) for 4 h at 37 °C. After that, the medium was removed, and 100 μl of DMSO was added to dissolve the formazan crystals. The absorbance at 570 nm was measured by an automated plate reader (Tecan Infinite M200 PRO, Switzerland).

**PAGE analysis.** The conjugation between MSN and the aptamer was determined by PAGE analysis. Briefly, 5 μl of all samples ((1) DNA ladder; (2) MSN; (3) Free aptamer; (4) MSN mixed with aptamers; (5) MSN-AP-; (6) MSN-AP; (7) MSN-AP− incubated with rat blood for 8 h; (8) MSN-AP incubated with rat blood for 8 h) were mixed with 2.6 μl of SYBR green and 7.5 μl of loading buffer, and incubated for 15 min at 25 °C followed by PAGE analysis. The results were analysed by using the Bio-Rad ChemiDoc MP system.

**Flow cytometry analysis of MSN-exosome conjugates.** Exosomes were attached to 2.7 μm aldehyde sulphate latex beads (Invitrogen) by mixing 10 μg of exosomes with 10 μl of beads for 15 min at room temperature with continuous rotation. This suspension was diluted to 1 ml with PBS and rotated for another 30 min. The reaction was stopped with the addition of 100 mM glycine and 2% BSA in PBS with 30 min of rotation. The exosome-bound beads were washed in PBS with 2% BSA and centrifuged at 15,000 g for 1 min. The beads were blocked with 10% BSA for 30 min with rotation, washed again in 2% BSA, centrifuged for 1 min at 15,000 g and incubated with MSN-AP-Cy− or MSN-AP-Cy (100 μg per ml) for 30 min with rotation at 4 °C. The beads were centrifuged at 15,000 g for 1 min, and the supernatant was discarded. The beads were washed in 2% BSA and centrifuged at 15,000 g for 1 min. The blank beads alone were used as a control. The percentage of exosome-bound beads was calculated relative to the total number of beads analysed per sample, and referred to as the percentage of beads bound to EGFR exosomes.

To examine the specific binding between MSN-AP- or MSN-AP and EGFR-expressing A-Exo from A549 cells (H-Exo from HELF cells was used as a control), we incubated A-Exo or H-Exo (30 μl, 40 μg per ml) with 10 μl of CD9-coated beads (2.7 μm), and 20 μl of either MSN-AP-Cy− or MSN-AP-Cy (100 μg per ml) for 30 min with rotation. The mixture was centrifuged at 15,000 g for 1 min, and the sediment was collected and washed twice with PBS containing 2% BSA. The resulting product was suspended in 500 μl of PBS and analysed by flow cytometry.

The ability of MSN-AP to capture the targeted exosomes in rat blood was examined. Rat blood was collected in an EDTA tube. Fifty-five and 550 ng of A-Exo were added to two 300 μl of blood samples. Then, 50 μl of 100 μg per mlMSN-AP-Cy− or MSN-AP-Cy was added to the mixture and incubated at 37 °C under both rocking and static conditions for 1 h. The resulting product was centrifuged at 200 g for 5 min, and the supernatant was centrifuged at 15,000 g for 10 min, and washed with PBS three times. MSN-Exo was then incubated with anti-CD9 beads for flow cytometry analysis.

To test the stability of MSN-Exo conjugates, 10 μl (40 μg per ml) of PKH67-labelled A-Exo and 50 μl (100 μg per ml) of MSN-AP-Cy were, respectively, added to 200 μl of rat blood, and mixed at 100 rpm for 4 h at 37 °C. The suspension was diluted to 1 ml with PBS and centrifuged at 200 g for 5 min to remove the deposits. The supernatant was centrifuged at 15,000 g for 10 min to collect the deposit (MSN-AP-Cy or MSN-Exo). The deposit was incubated with 10 μl of anti-CD9 beads, washed twice with PBS, and suspended in 500 μl of PBS to examine the formation of MSN-Exo and their 4 h stability in blood using flow cytometry. To determine whether MSN-AP can conjugate with blood exosomes, 50 μl (100 μg per ml) of MSN-AP-Cy was added to 200 μl of rat blood to be examined by flow cytometry.

**Confocal microscopy analysis of MSN-exosome conjugates.** The binding between MSN-AP and A-Exo or H-Exo was determined by confocal microscopy (Leica TCS SP8, Germany) analysis. In brief, PKH67-labelled exosomes (30 μl, 40 μg per ml) were incubated with MSN-AP-Cy− or MSN-AP-Cy (20 μl, 100 μg per ml) at 4 °C for 30 min. The resulting products were incubated with Hoechst-stained HELF cells for another 30 min followed by confocal microscopy examination based on co-localisation[10,39]. The interaction between cationized MSN and A-Exo was determined via a similar method.

To quantify the exosomes in blood and tissues, each sample was captured with five random visual fields at 63-fold magnification with confocal microscopy to count the number of exosomes. We first examined the stability of the formed MSN-Exo in blood for up to 4 h after incubation of PKH67-labelled A-Exo with MSN-AP-Cy in rat blood at 37 °C. The mixture was centrifuged at 200 g for 5 min to remove the deposits. The supernatant was centrifuged at 15,000 g for 10 min to collect the deposit (MSN-AP-Cy or MSN-Exo). The deposit was added to a confocal dish containing LO2 cells using a confocal microscope 30 min later. We then examined the stability of MSN-Exo in LO2 cells. PKH67-labelled A-Exo (10 μl, 40 μg per ml) was incubated with MSN-AP-Cy (30 μl, 40 μg per ml) at 37 °C for 1 h. The resulting products were centrifuged at 15,000 g for 10 min, washed twice with PBS and incubated with LO2 cells at 37 °C for 1 h. The LO2 cells were digested and applied to the top of a transwell membrane with an 8 -μm pore size. After 4 h of incubation at 37 °C, samples were collected from the lower chamber and added to a confocal dish with LO2 cells for confocal microscopy analysis.

**TEM analysis of MSN-exosome conjugates.** The binding between MSN-AP and A-Exo in cell medium and rat blood was characterised by TEM (HT7700, Hitachi, Japan) following centrifugation at 15,000 g and washing after 4 h of incubation at 37 °C. SEM was used to ensure the conjugation.

**Intermolecular forces.** AFM can be used to determine intramolecular and inter-molecular forces and nanometre distances between molecules[40]. A-Exo solution was immobilised and dried on silicon wafers overnight, followed by washing with H$_2$O. MSN-AP (10 μl, 40 μg per ml) was added to the silicon wafers, which remained stationary for 1 h. After MSN-AP and A-Exo were conjugated, the tip of the AFM cantilever contacted MSN-Exo and was then separated from MSN-Exo. The relationship between the intermolecular force and the separation distance was recorded as single-molecular force spectroscopy.

**Transcytosis analysis.** The ability of MSN-Exo to dynamically transverse different kinds of liver cell lines was analysed using transwells. Human LO2 hepatocyte, cholangiocyte HIBEpiC cells and HUVECs were incubated at 37 °C on the upper chamber of the transwell membrane at 50,000 cells per well to form monolayers. Kupffer cells were added to the LO2 monolayers or HUVEC monolayers at a ratio of 1:6 (Kupffer: LO2 or Kupffer: HUVECs). One hour after the co-incubation at 37 °C, the formed MSN-Exo and its precursors (30 μl of 100 μg per ml MSN-AP-Cy− and MSN-AP-Cy, or 30 μl of 40 μg per ml of PKH67-labelled exosome) were added to the upper chamber of the transwell. The medium in the lower chamber (200 μl) was collected after 1, 2 and 4 h to quantify the MSN-Exo passing through the cell monolayers.

To watch the on-time the cellular trafficking of the A-Exo alone or MSN-Exo, we programmed the confocal microscope time-lapse sequences to continuously record the trafficking of MSN-Exo and A-Exo alone when they were incubated with human hepatocytes on the confocal dishes for up to 600 min, and the results were observed under the confocal microscope with 63-fold magnification.

**MSN-Exo endocytosis and exocytosis.** To determine whether endocytosis and exocytosis are involved in MSN-Exo traversing hepatocytes, we cultured LO2 cells on a confocal dish in the presence of the endocytosis inhibitor Dynasore (50 μM) for 1 h at 37 °C. The cells were washed with PBS followed by the addition of PKH26-labelled MSN-Exo to the cell culture for 30 min. After washing the cells with PBS three times, the PKH26-labelled MSN-Exo signals generated from LO2 cells were analysed by confocal microscopy and flow cytometry. In addition, we incubated LO2 cells on a confocal dish in the presence of PKH26-labelled MSN-Exo for 30 min. After washing the excess MSN-Exo with PBS three times, the LO2 cells were incubated with the exocytosis inhibitor BafilomycinA1 (100 nM) for 6 h, and the PKH26-labelled MSN-Exo signals generated from LO2 cells were analysed by confocal microscope and flow cytometry and compared with PBS treatment only.

**Patient sample analysis.** Patient peripheral blood samples were obtained from non-small cell lung cancer patients of Fuzhou General Hospital. All individuals provided informed consent for blood analysis approved by the Institutional Review Board of the hospital (protocol #201705). The blood samples were collected into EDTA-containing tubes and used within 12 h after collection. Each blood sample was divided into three aliquots (500 μl each) and mixed with 50 μl of MSN-AP-Cy (40 μg per ml) to capture patient exosomes. The mixture was incubated at 37 °C for 1 h with 100-rpm rocking, and then centrifuged at 200 g for 5 min to remove the blood cells. The supernatant was centrifuged again at 15,000 g for 10 min, and the deposit was washed twice with PBS. The formed MSN-Exo was mixed with 10% BSA for 30 min with rocking followed by MSN-Exo quantification by flow cytometry analysis.

**Animal studies.** All animal studies were performed in accordance with animal protocol procedures approved by the Institutional Animal Care and Use Committee (IACUC) of Fuzhou University, which are consistent with AAALAS guidelines. All animals were monitored for abnormal behaviours to minimise animal pain and suffering. Animals were euthanized if excessive deterioration of animal health was noted.

To determine the effects of MSN-AP on the tissue distribution of the targeted exosomes in mice, we divided immunocompetent C57BL/6 mice into different groups ($n = 4$–5 per group) for comparison and analysis purposes. The mice were injected with PKH67-labelled H-Exo or A-Exo (both 10 μg via the tail vein) followed by intravenous administration of 200 μl of physiological saline, MSN or MSN-AP (both 5 mg per kg). The mice were killed 2 or 4 h later, and their organs (heart, liver, spleen, lung, kidney and small intestine) were collected and frozen-sectioned for exosome quantification by confocal microscopy analysis. To determine the effect of MSN-AP on A-Exo elimination, mice were injected with PKH67-A-Exo and MSN-AP-Cy, followed by tissue examination of MSN-Exo with a confocal microscope 2 or 4 h later.

To determine the types of cells that took up exosomes, PKH26-A-Exo was injected into C57BL/6 mice, and the organs were collected after 2 or 4 h. The organs were frozen-sectioned and stained with antibodies against F4/80 (1:50, Abcam, FITC, AB60343) and EpCAM (1:100, Invitrogen, FITC, 11–5791–82) for confocal imaging.

To determine the effects of MSN-AP on the kinetic changes of circulating exosomes, C57BL/6 mice were injected with PKH67-labelled H-Exo or A-Exo (both 10 µg via the tail vein) followed by intravenous administration of 200 µl of physiological saline, MSN, MSN-AP− or MSN-AP (all 5 mg per kg). Ten minutes, 2 h and 4 h later, mouse blood (200 µl) was drawn from the retro-orbital sinus and centrifuged at 200 g for 10 min to remove blood cells. The circulating exosomes were added to a confocal dish with LO2 cells and quantified by confocal microscopy. After single administration of MSN-AP, the blood total exosomes of tumour-bearing mice were extracted from 500 µl of mouse blood at 10 min, 2 h and 4 h with a blood extraction kit (UmibioScience and Technology). The total blood exosomes were labelled with PKH67 and added to a confocal dish with LO2 cells for confocal microscopy quantification.

To determine the effects of MSN-AP on hepatobiliary elimination of the circulating exosomes, we first determined whether MSN-AP could expedite hepatobiliary elimination of the targeted exosomes when the exosomes were given locally. Foreign DNA-transfected H-Exo or A-Exo (10 µg) was injected into the hepatic portal vein of SD rats after we ligated the upper and lower ends of the duodenum of anaesthetised rats. Ten minutes later, MSN-AP or saline (100 µl) was injected into the hepatic portal vein of the rats, and the rat duodenum contents were collected 4 h later and quantified for the levels of H-Exo and A-Exo using the qRT-PCR method. We then determined if MSN-AP could expedite hepatobiliary elimination of the targeted exosomes when the exosomes were given systemically. A-Exo (50 µg) was injected into SD rats through the tail vein followed by administration of saline or MSN-AP (200 µl, 5 mg per kg). We collected bile from the anaesthetised rats during the period of 0–100, 100–200, 200–300 and 300–400 min for quantification of the targeted exosomes using the qRT-PCR method based on the log standard curve.

To evaluate whether elimination by MSN-AP of the circulating oncogenic exosomes derived from A549 cells could attenuate A549-induced lung metastasis, we subcutaneously implanted A549 cells into female nude mice. Two weeks later, subcutaneous neoplasia occurred. The mice were divided into four groups (n = 4). Each group received intravenous saline, MSN, MSN-AP (both 5 mg per kg), or A-Exo (2 µg), every 3 days for 3 weeks. The mice were killed 7 weeks after the A549 implantation, and the metastatic nodules in their lungs were quantified and compared between the groups. The hearts, livers, spleens, lungs and kidneys of the mice were collected and fixed in 10% buffered neutral formalin, embedded in paraffin, sectioned and stained with haematoxylin and eosin for histological examination.

**Statistical analyses**. GraphPad Prism version 5.01 was used for statistical analyses. Data presented as the mean ± s.e.m. Statistical significance was determined using one-way ANOVA or unpaired two-tailed Student's t test. P < 0.05 was considered statistically significant.

**Reporting summary**. Further information on research design is available in the Nature Research Reporting Summary linked to this article.

## Data availability
All source data, associated code and additional results of this study are available from the corresponding author upon reasonable request.

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

## Acknowledgements

This work was supported by the Ministry of Science and Technology of China (2015CB931804), the National Natural Science Foundation of China (81773063; U1505225; 21907014; 81273548; 81571802; 81703555), the Natural Science Foundation of Fujian Province (2016J06020), and the China Postdoctoral Science Foundation (2017M620268). We thank Dr. Xinqi Zhang for performing the TEM experiments.

## Author contributions

L.J. conceptually designed the experimental strategy, provided intellectual input, supervised the project and wrote the paper. X.X. designed the detailed experiments, performed the experiments, analysed the data and helped to write the paper. H.N. and F.L. performed the cell culture and exosome isolation. H.M. performed the preparation of the nanoparticles. S.L. analysed the distribution of the exosomes. Y.Z., T.L. and B.L. performed the animal experiments. Y.L., H.D., J.W., M.L. and C.W. provided intellectual input and helped design the experimental strategy, J.S. and Y.G. prepared some figures. J.C. and F.X. provided clinical samples and recommendations.

## Competing interests

The authors declare no competing interests.
