## [Peer Review File · Nature Communications]

Reviewers' Comments:

Reviewer #1:

Remarks to the Author:

In this manuscript, Xie et al. developed mesoporous silica nanoparticle (MSN) modified with EGFR-specific aptamer (MSN-AP) in order to promote elimination of EGFR positive exosomes from blood circulation via hepatobiliary excretion. The presented concept is novel and of interest to some population of readers. However, although the concept is interesting, experimental design is not good enough to prove the concept. Moreover, although the reviewer can understand the novelty of the concept, the reviewer cannot understand the necessity and advantage of the concept. Followings are comments to general points and to specific points.

General points.

1. Lack in control MSN.

Considering that cationized MSN without aptamer can interact with exosomes that possess negative charge, cationized MSN should be included as a control group.

2. Data presentation.

Relative amount of exosomes such as arbitrary units and cycle numbers is not suitable. Rather, absolute amount of exosomes such as % of input exosomes should be used to show experimental data as measurement of fluorescent intensity and real-time PCR analysis can be used to calculate absolute amount of exosomes.

In addition, as weight of MSN provide less information compared to the number of MSN particles, the number of MSN per mg of MSN should be described.

3. The necessity and advantage of the concept

As demonstrated by experiment shown in Fig.4, most, if not all, of exosomes and exosomes interacted with MSN are mainly trapped by Kupffer cells. Therefore, exosomes are usually rapidly cleared from blood circulation. In addition, as clearance by hepatobiliary excretion is not different from clearance by Kupffer cells from the viewpoint to increase clearance of exosomes in blood, the reviewer cannot understand why the authors insisted on hepatobiliary excretion. For example, as shown in Fig.3, it seems that three MSNs seem to interacted with an exosome particle.

4. Evaluation of hepatobiliary excretion.

If exosomes are excreted to intestine via hepatobiliary excretion, exosomes should exist in the contents (or feces) in the intestine. On the other hand, exosome can distribute to the intestine not only via hepatobiliary excretion but also via blood circulation. Therefore, amount of exosomes in the contents of intestine scraped from the tissue should be evaluated.

Specific points

1. Page 5, second paragraph

"APTES" and "TEOS" should be spelled out.

2. Page 6, first paragraph: "The fluorophore and AP in MSN can be protected by the MSNs' highly ordered hexagonal pore structure and adjustable pore size (1.5-10 nm) 22."

If the aptamer is protected inside of pore of MSN, it cannot recognize EGFR. What does this description mean?

3. Page 6 second paragraph: "The number of aptamers on MSN were 0.041 and 0.043 nmol/mg, respectively, for MSN-AP-Cy3 and MSN-AP-Cy5 measured by fluorescent calibration curves of Cy3 or Cy5 (Figure S2 a-b)."

The number of aptamer per single MSN particle should be determined.

4. Page 10, first paragraph: " By analyzing the numbers of the transcytosed nanomaterials and their half-life in the presence and absence of Kupffer cells, we determined that more than 50 % of the nanomaterials were engulfed by Kupffer cells, and the rest could traverse across sinusoidal plasma membrane and canaliculae membrane into bile duct."

By comparing data shown in Fig. 3g (LO and Kupffer cells, less than 0.8% for MSN-Exo at 4h) and 3k(LO cells alone, approximately 16% for MSN-Exo at 4h), the reviewer considers that more than 95 % of the nanomaterials were engulfed by Kupffer cells.

5. Page 10, second paragraph: "Bafilomycin A1 is an exocytosis inhibitor for phosphonate-modified MSN29. Dynasore treatment significantly reduced intracellular MSN-Exo by 4-fold in comparison with PBS treatment evidenced by both flow cytometry (Figure 4m, n) and confocal microscopy (Figure S 16). Bafilomycin A1 treatment significantly enhanced intracellular MSN-Exo by 1.2-fold in comparison with PBS treatment shown by flow cytometry (Figure 4o, p) and confocal microscopy (Figure S 16)."

Bafilomycin A1 is an inhibitor of endosome acidification and treatment of Bafilomycin A1 inhibit endosomal degradation. Therefore, increase in signal in the cells does not directly indicate decrease in transcytosis. If the authors would like to claim that Bafilomycin A1 treatment decreased transcytosis, the number of exosomes that passed through the cells must be evaluated.

6. Page 15: About "exosome transfection kit (Weihui bio, Beijing)"

The reviewer cannot find this reagent. Is this Exo-Fect Exosome Transfection Kit from system biosciences?

7. Fig2: About the existence of aptamer on MSN.

Direct detection of aptamer on MSN by TEM is desirable.

8. Fig.2h and i

Not only magnified images, but also decreased image that show wider field must be shown. In these pictures, vesicles that are bound to MSN (that can be recognized by TEM) are indicated by red arrows as A-Exo. However, there are abundant amount of exosomes in serum. Therefore, even if it seems that a vesicle bound to MSN, it is not impossible to distinguish whether the vesicle is A-Exo mixed with serum or exosome in the serum.

8. Fig.4

As exosomes alone are easily taken up by cells, not only exosomes mixed with MSNs, but also exosomes alone should be included.

9. Fig 5d and Figs S17 and S18.

As the types of cells that took up exosomes are important, identification of the cells that took up exosomes by immune-staining should be performed.

10. Fig. 6c and d.

Not only the effect of MSN-AP administration on metastasis, but also the its effect on blood exosomes must be investigated. At least, time-course of the amounts of total exosomes and EGFR-positive exosomes in the blood before and after the single administration of MSNs must be investigated in tumor-bearing mice.

Reviewer #2:

Remarks to the Author:

The authors reported the design of EGFR-targeting aptamer modified mesoporous silica nanoparticle (MSN-AP) to eliminate the cancer-causing exosome/ disease-relative biohazard from the blood, and tow the A-Exo across hepatobiliary layers and Oddi's sphincter into small intestine. In vitro data showed good recognition and binding between MSN-AP and A-Exo and in vivo results found A-Exo were able to clean from the blood in the present of MSN-AP. However, some experiment results do not support the conclusion, the description for the results not clear enough and several typos were found in the manuscript. In addition, the manuscript need professional English editing for better understand. The quality of the manuscript needs to be improved before considering for publication. The authors should revise this manuscript to address the concerns given below:

1. In this study, the author did not describe the reason for using mesoporous silica nanoparticle (MSN). Nano-channels of MSN were not needed in this work.
2. More detail nanoparticle characterization should be done. For example TGA or FTIR to confirm the EGFR aptamer conjugation.
3. In vitro MSN-Exo conjugation assay (Figure 3a), did MSN-AP conjugate with A-Exo first and then endocytosis into LO2 cell or both MSN-AP and A-Exo entered LO2 cells and then conjugated? It's important to verify the sequence since author claimed MSN-AP help to clean out the A-Exo in the blood and eliminate. And please explain why the H-Exo was found in the cell without MSN-AP/MSN-AP- (Figure 3a).
4. What is the rocking/static condition stand for in physiological environment?
5. In figure 4(a), flow cytometry should monitor both channels for Cy5 and PKH67.
6. In Figure 4(c), author indicated the biostability of MSN-exo after 4 hrs incubated inside LO2 cell. On the other hand, Figure 4(d) and (e) showed the MSN-Exo excreted out of LO2 cells within 20 mins. Author should explain more about the conflict of results.
7. In Figure S5, yellow color of MSN-Exo- were found inside of cell at early time point (30 to 105 mins), and then those dots became green color at later time point (230 to 600 mins). Does it imply that MSN-AP- dissociate with A-Exo? MSN-AP- excreted out of Kuffer cell without A-exo?
8. The results do not match to the conclusion. In page 9, line 6, author would like to further demonstrate that endocytosed MSN-Exo can be excreted by LO2 as an entity but showed the result of red dots (representing MSN-AP) were found in figure S4a, what does this means?
9. How exosome labeled with PKH67? Covalent bond or electrostatic interaction? Since several important images were monitored the PKH67 of A-Exo, we should further confirm the signal the authors observed in the images were A-Exo with PKH67 instead of PKH67 alone. The labeling detail should be address in the exosome labeling process.
10. Figure S11-15 showed the cell images from the transwell membrane, are those cells from the upper chamber? If yes, the fluorescent intensity of MSN-Exo or MSN-AP(+/-) should gradually decrease as a function of time (1, 2, 4 hours) since the nanoparticle have transverse to lower chamber. Author should explain why the fluorescence intensity increases in Figure S11-15.
11. Figure 5a, the increase amount of A-Exo and H-Exo in the rat duodenum content after MSN-AP was similar, no significant difference.
12. For in vivo study, most of the images were relied on the PKH67 of A-Exo. Author should also include the MSN-AP bio-distribution for in vivo kinetic distribution study to prove the in vivo co-localization of A-Exo and MSN-AP such as monitoring the Cy5 fluorecence from IVIS image system or ICP-MS for silica contents.
13. Figure 5d and Figure S17 and S18 should also include the bio-distribution of MSN-AP to prove the movement of A-Exo was helped by MSN-AP.
14. In figure 3e, positive charge MSN-AP showed significant higher A-Exo binding efficiency (~100

times) than negative charge MSN-AP-, but showed equally amount of blood exosome clearance efficacy in Figure 5c (2h and 4h). Author should explain for detail and incorporate in the manuscript.

15. In the Figure caption of Figure S10, "Trafficking of the positively-charged MSN-Exo- in and out of the same cholangiocyte." Does it means "negative -charged"?

16. Page 9, line 11 described "The confocal microscopic analysis (Figure 4c) showed the merged yellow dots from the MSN-Exo excreted from the LO2 as an entity." The yellow dots in figure 4c still inside of cell. It should be a typo. The authors should check the description and the figures throughout the manuscript.

Lee Jia, Ph.D. FAAPS.
Director, Cancer Metastasis Alert and Prevention
Center
Distinguished Professor
College of Chemistry, Fuzhou University
Sunlight Building, 6FL, Science Park, Xueyuan
Road, University Town;
Fujian 350116, China
E-mail:pharmlink@gmail.com
cmapcja1234@163.com

August 1, 2019

Re: NCOMMS-19-09320-T

Dr. Kyle Legate
Senior Editor
Nature Communications

Dear Dr. Legate,

Thank you for providing us with a clear guidance about what we need to do to improve and revise our manuscript for publication in your journal. We find the reviewers' comments useful, and did major revisions to address their criticisms. We have revised our manuscript according to the reviewers' comments, and corrected the formats and errors. To satisfy the referees and answer their questions with data, we added 12 Supplementary Figures that are Suppl. Fig. 3, 4, 6, 7, 8, 22, 23, 24, 25, 28, 29 and 30. These changes are highlighted **in red** in the revised manuscript for your convenience.

The following are our point-by-point responses **in blue** to the comments and suggestions of the reviewers:

Reviewer #1 (Remarks to the Author):

In this manuscript, Xie et al. developed mesoporous silica nanoparticle (MSN) modified with EGFR-specific aptamer (MSN-AP) in order to promote elimination of EGFR positive exosomes from blood circulation via hepatobiliary excretion. The presented concept is novel and of interest to some population of readers. However, although the concept is interesting,

experimental design is not good enough to prove the concept. Moreover, although the reviewer can understand the novelty of the concept, the reviewer cannot understand the necessity and advantage of the concept. Followings are comments to general points and to specific points.

General points.

1. Lack in control MSN.

Considering that cationized MSN without aptamer can interact with exosomes that possess negative charge, cationized MSN should be included as a control group.

Response: A very good point! Thanks. We conducted an experiment using cationized MSN without aptamers conjugated to it and incubating the cationized MSN with PKH67-labeled A-Exo for 1h. After centrifugation, the precipitated products were examined by confocal microscopy. The additional experiment shows no significant specific binding between the cationized MSN and A-Exo. The related experimental results are showed together in **Supplementary Fig. 4** for your review with word descriptions in both the legends of Figure 3 and 4, and in the text.

2. Data presentation.

Relative amount of exosomes such as arbitrary units and cycle numbers is not suitable. Rather, absolute amount of exosomes such as % of input exosomes should be used to show experimental data as measurement of fluorescent intensity and real-time PCR analysis can be used to calculate absolute amount of exosomes.

In addition, as weight of MSN provide less information compared to the number of MSN particles, the number of MSN per mg of MSN should be described.

Response: We have changed the data presentation of exosome amount from the relative amount to the absolute amount as you required, and presented the absolute amount of exosomes in Figure 5 and Figure 6b.

We used Nanoparticle Tracking Analysis (NTA) method, and determined that per mg of MSN contains 7×10^{10} MSN nanoparticles calculated after measuring MSN amount in a part of MSN standard solution by the NTA method.

3. The necessity and advantage of the concept

As demonstrated by experiment shown in Fig.4, most, if not all, of exosomes and exosomes interacted with MSN are mainly trapped by Kupffer cells. Therefore, exosomes are usually rapidly cleared from blood circulation. In addition, as clearance by hepatobiliary excretion is not different from clearance by Kupffer cells from the viewpoint to increase clearance of exosomes in blood, the reviewer cannot understand why the authors insisted on hepatobiliary excretion. For example, as shown in Fig.3, it seems that three MSNs seem to interacted with an exosome particle.

Response: Thank you for your positive comment on the necessity and advantage of the

concept that we presented in the manuscript. After PubMed searching, we have not found a paper that clearly describes a technology developed for eliminating blood biohazards into small intestine. The present study creates a novel avenue that uses the intelligently-designed biosafe MSN to precisely identify, tow and dump the circulating biohazards (cancer exosomes are an example of the biohazards) into small intestine. Although Figure 4g-4h show most of exosomes bound to the MSN are mainly trapped by Kupffer cells, the results from the in vitro transwell model has to be correlated with the results from the in vivo models, and the latter simulates the true situation of hepatobiliary excretion. The route of hepatobiliary excretion of a circulating molecule is usually through the following pathway: 1) liver sinusoid, where Kupffer cells reside; 2) space of Disse; 3) hepatocytes; 4) bile ducts, and 5) intestines. Although Kupffer trap (or Kupffer cell phagocytic response) is the first and major barrier to nanomaterials (Poon, et al. ACS Nano 2019; 13: 5785-98), Kupffer clearance does not represent the whole hepatobiliary excretion system. Each step in the above route could influence hepatobiliary excretion rate of a circulating molecule and its elimination T1/2 in the body. In addition, the size, core density, surface charge, and surface chemistry of the excreted molecules also influence their hepatobiliary excretion rate (see our publication: Ref 1). Kupffer cells are the specialized stellate macrophages in the liver, their key physiological function is to phagocytize pathogens and aged erythrocytes.

4. Evaluation of hepatobiliary excretion.

If exosomes are excreted to intestine via hepatobiliary excretion, exosomes should exist in the contents (or feces) in the intestine. On the other hand, exosome can distribute to the intestine not only via hepatobiliary excretion but also via blood circulation. Therefore, amount of exosomes in the contents of intestine scraped from the tissue should be evaluated.

Response: We analyzed amount of exosomes in both the blood (Figure 5c) and the contents of small intestine (Figure 5d-e, the rightmost panels), and in other major organs. In addition, our quantitative analysis (Figure 6b) shows that the dynamic decrease of blood exosomes was sequentially accompanied by an increase of exosomes in small intestine.

General response: This reviewer not only asked us many challenging questions, but also provided us with many constructive suggestions to improve the quality of our manuscript. This reviewer appreciates the novelty of our study. To answer your questions with new data, we added 12 Supplementary Figures that are Suppl. Fig. 3, 4, 6, 7, 8, 22, 23, 24, 25, 28, 29 and 30. We hope that our responses could satisfy you. For all of these, we sincerely thank this reviewer for supporting our work.

Specific points

1. Page 5, second paragraph
“APTES” and “TEOS” should be spelled out.

Response: Thank you very much for telling us where the abbreviations are that should be fully spelled out when they appear in the text for the first time. We made a change in writing as “Aminopropyltriethoxysilane (APTES) and tetraethyl orthosilicate (TEOS)” in the same place.

2. Page 6, first paragraph: “The fluorophore and AP in MSN can be protected by the MSNs’ highly ordered hexagonal pore structure and adjustable pore size (1.5-10 nm) 22.”

If the aptamer is protected inside of pore of MSN, it cannot recognize EGFR. What does this description mean?

Response: Your question makes a sense. To be rigorous in writing, we rewrote the sentence (first paragraph of page 6) as follows: “The partial fluorophore and AP in MSN can be protected by the MSNs’ highly ordered hexagonal pore structure and adjustable pore size (1.5-10 nm)”. On the other hand, there are many EGFR-targeting AP on the surface of the MSN, which can recognize EGFR.

3. Page 6 second paragraph: “The number of aptamers on MSN were 0.041 and 0.043 nmol/mg, respectively, for MSN-AP-Cy3 and MSN-AP-Cy5 measured by fluorescent calibration curves of Cy3 or Cy5 (Supplementary Fig 2 a-b).”

The number of aptamer per single MSN particle should be determined.

Response: As you pointed out that (the above question 2) the number of MSN per mg of MSN should be described, and the absolute amount of exosomes should be described, we thus determined the number of aptamer per single MSN, as you required, to be 344 aptamers per single MSN, and added the information to the second paragraph of page 6 of the text as follows: “The number of aptamers on MSN were 0.041 and 0.043 nmol/mg, respectively, for MSN-AP-Cy3 and MSN-AP-Cy5 measured by fluorescent calibration curves of Cy3 or Cy5 (**Supplementary Fig 2 a-b**). Based on the information, we determined that per single MSN particle contains 344 aptamers.”

4. Page 10, first paragraph:” By analyzing the numbers of the transcytosed nanomaterials and their half-life in the presence and absence of Kupffer cells, we determined that more than 50 % of the nanomaterials were engulfed by Kupffer cells, and the rest could traverse across sinusoidal plasma membrane and canniculae membrane into bile duct.”

By comparing data shown in Fig. 3g (LO and Kupffer cells, less than 0.8% for MSN-Exo at 4h) and 3k(LO cells alone, approximately 16% for MSN-Exo at 4h), the reviewer considers that more than 95% of the nanomaterials were engulfed by Kupffer cells.

Response: Your consideration sounds reasonable. Although Figure 4g-4h show that most of MSN-Exo were mainly trapped by Kupffer cells that were co-cultured with a monolayer of either LO2 or endothelial cells on the transwell, the in vitro model has to be correlated with the in vivo models, and the latter simulate the true situation of hepatobiliary excretion. It is reported that 94% of endocytosed MSN could be exocytosed by the cells and detected in the cell media 24 h later (Small 2013; 9: 697-704). In mice and rats, within 10 min of tail-vein

injection of MSN, nearly all of the injected MSN was sequestered by the liver, and 30 min later a good percentage of MSN began to migrate into the GI tract. By 4 h, the overwhelming majority of MSN had concentrated within the jejunum and duodenum (Biomaterials 2010; 31: 5564-74).

Having said that, we accepted your suggestion, and changed the sentences as follows (last two sentences on page 10, first paragraph): “By analyzing the numbers of the transcytosed nanomaterials and their half-life in the presence and absence of Kupffer cells, we determined that more than 95% of the nanomaterials were engulfed by Kupffer/LO2 or Kupffer/endothelial co-incubated monolayers, and the rest could traverse across sinusoidal plasma membrane and canaliculae membrane into bile duct. However, the Kupffer/LO2 and Kupffer/endothelial co-incubation models may not truly reflect the in vivo models.”

We hope that the revisions could satisfy you.

5. Page 10, second paragraph: “Bafilomycin A1 is an exocytosis inhibitor for phosphonate-modified MSN29. Dynasore treatment significantly reduced intracellular MSN-Exo by 4-fold in comparison with PBS treatment evidenced by both flow cytometry (Figure 4m, n) and confocal microscopy (Figure S 16). Bafilomycin A1 treatment significantly enhanced intracellular MSN-Exo by 1.2-fold in comparison with PBS treatment shown by flow cytometry (Figure 4o, p) and confocal microscopy (Figure S 16).”

Bafilomycin A1 is an inhibitor of endosome acidification and treatment of Bafilomycin A1 inhibit endosomal degradation. Therefore, increase in signal in the cells does not directly indicate decrease in transcytosis. If the authors would like to claim that Bafilomycin A1 treatment decreased transcytosis, the number of exosomes that passed through the cells must be evaluated.

Response: You are right. Bafilomycin A1 has been demonstrated as an inhibitor of both endosomal acidification and exocytosis (Small 2013; 9: 697-704). To satisfy you, we evaluated the number of exosomes that passed through the LO2 cells by performing an experiment in which, LO2 cells were incubated with PKH67-labeled MSN-Exo for 30 min followed by washing the cells with PBS to discard the free PKH67-labeled MSN-Exo. The LO2 cells were treated with Bafilomycin A1 for 6 h. The number of transcytosed PKH67-labeled MSN-Exo in the culture medium was quantified by confocal microscopy. The experiment result and word description can be found in the **Supplementary Fig. 22** We added the following sentence in the revised text, page 10, last paragraph, to claim that “Bafilomycin A1 treatment significantly decreased the exocytosed MSN-Exo”.

We hope that the revisions could satisfy you.

6. Page 15: About “exosome transfection kit (Weihui bio, Beijing)”

The reviewer cannot find this reagent. Is this Exo-Fect Exosome Transfection Kit from system biosciences?

Response: We brought the kit from Weihui bio, Inc., Beijing. After your inquiry, we contacted the biotech company, and found out that the product is manufactured by 101 Bio (California, USA) with its commercial name as "ExoFectin® sRNA-into-Exosome Kit (Chemical)." The

kit's catalog # is P401, and its original information can be found at <http://www.101bio.com/P401.php>

7. Fig 2: About the existence of aptamer on MSN.
Direct detection of aptamer on MSN by TEM is desirable.

Response: To meet your demand, we tried very hard to obtain the TEM images to show the aptamers on MSN by TEM but have not been lucky because the high voltage of TEM can interfere with high contrast resolution between MSN and aptamers or antibodies (Small 2016, 12: 2595–2608). However, the existence of aptamer on MSN has been proven by Figure 2i, where the PAGE electrophoresis shows lane 3 is free aptamers; lane 4 is MSN mixed with free aptamers; lane 5 is MSN-aptamers-; lane 6 is MSN-aptamer+. To satisfy you, we performed an additional FTIR experiment to demonstrate the existence of aptamers on MSN (**Supplementary Fig. 3**).

We hope that the PAGE electrophoresis image and FTIR scanning provide the enough evidence to prove the existence of aptamers on MSN.

8. Fig.2h and i

Not only magnified images, but also decreased image that show wider field must be shown. In these pictures, vesicles that are bound to MSN (that can be recognized by TEM) are indicated by red arrows as A-Exo. However, there are abundant amount of exosomes in serum. Therefore, even if it seems that a vesicle bound to MSN, it is not impossible to distinguish whether the vesicle is A-Exo mixed with serum or exosome in the serum.

Response: We guessed that you actually meant Fig.3h and i, instead of Fig.2h and i.

1) We have provided a SEM image (**Supplementary Fig. 6**) with the wide-field view to show A-Exo bound to the MSN-AP, and to distinguish A-Exo from exosomes in blood by incubating MSN-AP with blood that contains both A-Exo and other vesicles. After 4-h incubation under static or rocking condition, the samples were centrifuged, and the precipitate containing MSN-AP and MSN-AP bound to A-Exo was analyzed by SEM.

2) We provided you with one more **Supplementary Fig. 7** to show that the MSN-AP binds to A-Exo only, not to the normal exosomes in the blood by using flow cytometry.

8. Fig.4

As exosomes alone are easily taken up by cells, not only exosomes mixed with MSNs, but also exosomes alone should be included.

Response: We are delighted to accept your suggestion, and added an experiment to show uptake of PKH67-labeled exosomes by LO2 cells under a confocal microscope. We added the image as **Supplementary Fig. 8** with word description in the text of page 9 line 7 as follows: LO2 cells took up exosomes easily.

9. Fig 5d and Figs S17 and S18.

As the types of cells that took up exosomes are important, identification of the cells that took

up exosomes by immune-staining should be performed.

Response: We accepted your suggestion, and conducted additional experiments by injecting mice with PKH26-labeled A-Exo (red; 10 μ g via tail vein) followed by intravenous administration of MSN-AP (5 mg/kg), and identifying the types of the cells in major organs that predominantly take up exosomes by immune-staining 4-h later after the administration. All tissues were labeled with Hoechst, anti-F4/80 (green; for macrophages), or anti-EpCAM (green; for epithelial cells). If the tissues take up red PKH26-labeled A-Exo, we will see yellow dots. We found that liver and lung cells are the major types of cells that take up A-Exo. We presented the data in **Supplementary Fig. 25** for your review.

10. Fig. 6c and d.

Not only the effect of MSN-AP administration on metastasis, but also the its effect on blood exosomes must be investigated. At least, time-course of the amounts of total exosomes and EGFR-positive exosomes in the blood before and after the single administration of MSNs must be investigated in tumor-bearing mice.

Response: Your point is well taken. Following your suggestion, we conducted an additional experiment to show the time-course of the amounts of total exosomes and EGFR-positive exosomes (A-Exo) in the mouse blood before and after the single administration of MSN-AP to the tumor-bearing mice. The result was presented as **Supplementary Fig. 24** for your review.

Reviewer #2 (Remarks to the Author):

The authors reported the design of EGFR-targeting aptamer modified mesoporous silica nanoparticle (MSN-AP) to eliminate the cancer-causing exosome/ disease-relative biohazard from the blood, and tow the A-Exo across hepatobiliary layers and Oddi's sphincter into small intestine. In vitro data showed good recognition and binding between MSN-AP and A-Exo and in vivo results found A-Exo were able to clean from the blood in the present of MSN-AP. However, some experiment results do not support the conclusion, the description for the results not clear enough and several typos were found in the manuscript. In addition, the manuscript need professional English editing for better understand. The quality of the manuscript needs to be improved before considering for publication. The authors should revise this manuscript to address the concerns given below:

General Response: We thank you very much for your constructive suggestions and comments for improving the quality of this manuscript. To answer your questions with new data, we added 12 Supplementary Figures that are Suppl. Fig. 3, 4, 6, 7, 8, 22, 23, 24, 25, 28, 29 and 30. We are very grateful to you for your supports.

1. In this study, the author did not describe the reason for using mesoporous silica nanoparticle (MSN). Nano-channels of MSN were not needed in this work.

Response: Thank you for your above constructive comments. We will improve the quality of our manuscript with great diligence.

Regarding this question, we did describe the reason for choosing MSN on page 4, line 6-10, and in Discussion, the second paragraph.

2. More detail nanoparticle characterization should be done. For example TGA or FTIR to confirm the EGFR aptamer conjugation.

Response: we accepted your suggestion, and supplemented the FTIR data to verify the conjugation of EGFR aptamers to MSN (see **Supplementary Fig. 3** of the revised manuscript).

3. In vitro MSN-Exo conjugation assay (Figure 3a), did MSN-AP conjugate with A-Exo first and then endocytosis into LO2 cell or both MSN-AP and A-Exo entered LO2 cells and then conjugated? It's important to verify the sequence since author claimed MSN-AP help to clean out the A-Exo in the blood and eliminate. And please explain why the H-Exo was found in the cell without MSN-AP/MSN-AP- (Figure 3a).

Response: Figure 3a was the first test after engineering MSN-AP to see if MSN-AP (compared with MSN-AP-) could specifically recognize and bind to the target A-Exo (compared with H-Exo) in cell medium. MSN-AP conjugate with A-Exo first and then the formed MSN-Exo are endocytosed into LO2 cells. We described the bind in the middle of page 7.

It is important to verify the sequence about how MSN-AP recognizes the target A-Exo. In Figure 3b, 3d, and 3f, we showed that MSN-AP could specifically recognize and bind to the target A-Exo to form MSN-Exo in cell medium, and in the blood (Figure 3e, 3h, 3i). In parallel, we used H-Exo prepared from normal lung HELF cells without the EGFR biomarker (Figure 1f, far right) as the controls to demonstrate low binding specificity between H-Exo and MSN-AP in the cell medium (Figure 3a and 3c).

Figure 3a actually compares the binding between MSN-AP- and A-Exo or H-Exo in cell medium. It shows that there was almost no binding between MSN-AP- and H-Exo because the latter lacks EGFR. By comparison, there was good binding between MSN-AP- and A-Exo because A-Exo expresses high EGFR. Moreover, MSN-AP+ recognizes and binds to A-Exo more than MSN-AP- does. We explained why there was no significant binding between H-Exo and MSN-AP in the legend of Figure 3.

We hope that our explanations could satisfy you.

4. What is the rocking/static conditions stand for in physiological environment?

Response: Our rocking and static conditions were set in rat blood at 37⁰C. Different from static culture condition, the rocking culture condition tilts between angles of $\pm 12^\circ$ at a rate of 100 rpm, resulting in gravity-induced medium flow that periodically changes direction and hydrostatic pressure. The rocking culture condition increases opportunities for gas and

metabolite exchange, and more importantly, for more collisions and resulting binding between biomaterials (for example, MSN-AP) and their targets (for example, exosomes). We incubated MSN-AP and A-Exo in blood at 37°C, and rocked the mixture for 4 h to see if MSN-AP can catch more A-Exo under rocking than under static condition. The results demonstrate that MSN-AP can catch more A-Exo under rocking condition than under static condition (Figure 3e). More information about the rocking platform can be found in *Lab on a Chip* 2010, 10: 446-455; or 2015, 15:2269-77. Nonetheless, the in vitro condition cannot completely simulate the in vivo one.

5. In figure 4(a), flow cytometry should monitor both channels for Cy5 and PHK67.

Response: In figure 4a we used Cy5-labeled MSN-AP alone as the control for the same Cy5-labeled MSN-AP incubated with A-Exo for 4 h. The crest represents the MSN-Exo formed after the incubation. Nonetheless, to satisfy you, we supplemented an experiment, in which Cy5-labeled MSN-AP was incubated with PKH67-labeled A-Exo in rat blood for 4 h, and the result products were monitored using two channels of the flow cytometry for Cy5 and PHK67. The image was provided in **Fig. 4a** (revised) for your review.

6. In Figure 4(c), author indicated the biostability of MSN-exo after 4 hrs incubated inside LO2 cell. On the other hand, Figure 4(d) and (e) showed the MSN-Exo excreted out of LO2 cells within 20 mins. Author should explain more about the conflict of results.

Response: Figure 4(c) and Figure 4(d)/(e) came from different samples, respectively: Figure 4(c) demonstrates that the endocytosed MSN-Exo by LO2 can be excreted by LO2 as an entity. To demonstrate the free transcytosis of MSN-Exo, we incubated MSN-AP with A-Exo and centrifuged (at low rate of 15,000 g) the above incubation mixture. The pellet (MSN-AP or MSN-Exo) was incubated with LO2 cells. After washing with PBS and centrifuging at 200 g, we collected the LO2 cells that contained MSN-Exo and/or MSN-AP only. We incubated the LO2 cells on the transwell membrane, and collected the cell medium from the lower chamber of the transwell. The confocal microscopic analysis (Figure 4c) showed the merged yellow dots that should be the MSN-Exo excreted from the LO2 as an entity.

Figure 4(d) and 4(e) show 1) negative MSN-AP- (4d) forms less MSN-Exo than positive MSN-AP (4e) does; and 2) the free endocytosis and transcytosis of the same MSN-Exo occurred within the same LO2 cell recorded by the sequential time lapse images.

We did some revisions on page 9.

7. In Figure S5, yellow color of MSN-Exo- were found inside of cell at early time point (30 to 105 mins), and then those dots became green color at later time point (230 to 600 mins). Does it imply that MSN-AP- dissociate with A-Exo? MSN-AP- excreted out of Kuffer cell without A-exo?

Response: Yes, it does as you pointed out. Kupffer cells are the specialized stellate macrophages in the liver, their key physiological function is to phagocytize pathogens and

aged erythrocytes. These cells secrete chemokines and cytokines to participate in inflammatory reaction. The color change may also be related to fluorescence degradation after 4h.

8. The results do not match to the conclusion. In page 9, line 6, author would like to further demonstrate that endocytosed MSN-Exo can be excreted by LO2 as an entity but showed the result of red dots (representing MSN-AP) were found in figure S4a, what does this means?

Response: A good question, and this question is also related to the above question 6. On page 9, paragraph 3, we stated that we wanted to further demonstrate that the endocytosed MSN-Exo by LO2 cells can be excreted by LO2 cells as an entity. We therefore incubated MSN-AP and A-Exo, and centrifuged (at a low rate of 15,000 g) the incubation mixture. The collected precipitate (MSN-Exo and MSN-AP) was incubated with LO2 cells. LO2 cells were centrifuged (at a low rate of 200 g) and collected the precipitate that was the LO2 cells containing MSN-Exo and/or MSN-AP. We incubated the LO2 cells on the transwell membrane, and collected the cell medium from the lower chamber of the transwell. The confocal microscopic analysis (Figure 4c) showed the merged yellow dots that should be the MSN-Exo excreted from the LO2 as an entity. In addition, as shown in **Supplementary Fig. 9a** and you pointed out here, the LO2 cells also endocytosed the unreacted MSN-AP (red).

9. How exosome labeled with PKH67? Covalent bond or electrostatic interaction? Since several important images were monitored the PKH67 of A-Exo, we should further confirm the signal the authors observed in the images were A-Exo with PKH67 instead of PKH67 alone. The labeling detail should be address in the exosome labeling process.

Response: The PKH67 labeling technology is a proprietary membrane labeling technology to stably incorporate a green fluorescent dye with long aliphatic tails (PKH67) into lipid regions of the cell membrane (Wallace, et al. Cytometry 2008; 73A:1019-1034; Horan, et al. Meth. Cell Biol. 1990; 33: 469-490) by intermolecular interaction force (a hydrophobic force). For exosome labeling, we added the following paragraph to Methods section (page 16, last paragraph): For exosome labeling, we mixed exosomes with Diluent C reagent (Sigma-Aldrich; both 20 μ l). One- μ l of PKH67 was mixed with another 40 μ l of Diluent C reagent. The above two solutions were mixed and incubated for 5 min, followed by adding an equal volume of 1% BSA to stop the staining. The labeled exosomes were washed with PBS, and collected by ultracentrifugation.

As stated above, since we used BSA to stop the staining, and washed out the free PKH67 followed by ultracentrifugation to collect the PKH67-labeled exosomes, it is unlikely that there were free PKH67 left over. In another word, the images that we observed should come from PKH67-labeled exosomes only, not from the free PKH67.

10. Figure S11-15 showed the cell images from the transwell membrane, are those cells from the upper chamber? If yes, the fluorescent intensity of MSN-Exo or MSN-AP(+/-) should gradually decrease as a function of time (1, 2, 4 hours) since the nanoparticle have

transverse to lower chamber. Author should explain why the fluorescence intensity increases in Figure S11-15.

Response: Your understanding of the MSN-Exo across transwell membrane and the related dynamic change in fluorescence intensity is right. However, we collected the culture medium from the lower chamber (not from the upper chamber) as illustrated in Figure 4f, and added the medium containing MSN-Exo or MSN-AP(+/-) to a new confocal microscopic dish with LO2 cells for microscopic analysis (termed co-localization; see *Nature Communications* 2016; DOI: 10.1038/ncomms13588). Therefore, the fluorescent intensity of MSN-Exo or MSN-AP(+/-) from lower chamber gradually increases as a function of time. The presence of the uniformly-growing LO2 cells on the confocal microscopic dish provides a good contrast for both qualitative and quantitative analyses of nanomaterials. Consistent with quantitative data shown in Figure 4g-4k, **Supplementary Fig. 16-20** show less MSN-Exo across Kupffer/LO2 (**Supplementary Fig. 16**) or Kupffer/endothelial (**Supplementary Fig. 17**) monolayers than LO2 (**Supplementary Fig. 18**) or endothelial monolayers (**Supplementary Fig. 20**). We have revised the legends of **Supplementary Fig. 16-20** for easier understanding.

11. Figure 5a, the increase amount of A-Exo and H-Exo in the rat duodenum content after MSN-AP was similar, no significant difference.

Response: This experiment was done by a single portal-vein injection of A-Exo or H-Exo (both 50 μg) to the rats, who were pretreated with or without MSN-AP injections (5 mg/kg). Anatomically-speaking, portal vein is very close to duodenum (Figure 6a; also referring to *ACS Nano* 2019; 13: 5785-98; *Drug Discovery Today* 2013; 19:326-40). We collected the intestine content for A-Exo or H-Exo analysis at 10 min after the local injections of A-Exo or H-Exo. The short distance between A-Exo or H-Exo injection site and duodenum, as well as the interference of the intestine content, the analysis is difficult. We carefully re-examined the original data, and used the absolute exosome amount, as required by another reviewer, to show excreted A-Exo or H-Exo in rat small intestine with or without local injections of MSN-AP. The revised Figure 5a is constructed based on the new standard curve (**Supplementary Fig. 23**) of exosome amount-PCR readout relationship. The revised Figure 5a is presented here for your review and the original data are submitted separately to Nature Communication.

12. For in vivo study, most of the images were relied on the PHK67 of A-Exo. Author should also include the MSN-AP bio-distribution for in vivo kinetic distribution study to prove the in vivo co-localization of A-Exo and MSN-AP such as monitoring the Cy5 fluorescence from IVIS image system or ICP-MS for silica contents.

Response: We respectfully accepted your suggestion. We did additional experiment to prove the in vivo co-localization of A-Exo and MSN-AP in major tissues. The **Supplementary Fig. 28-30** includes both PHK67-labeled A-Exo (green) and Cy5-labeled MSN-AP (red). We added these statements to the middle of page 12 of the manuscript text.

13. Figure 5d and Figure S17 and S18 should also include the bio-distribution of MSN-AP to prove the movement of A-Exo was helped by MSN-AP.

Response: We respectfully accepted your suggestion. We repeated Figure 5d experiment by using both PHK67-labeled A-Exo (green) and Cy5-labeled MSN-AP (red) to prove the movement of A-Exo was helped by MSN-AP. The new data are provided in **Supplementary Fig. 28-30**. We made changes in the corresponding figure legends.

14. In figure 3e, positive charge MSN-AP showed significant higher A-Exo binding efficiency (~100 times) than negative charge MSN-AP-, but showed equally amount of blood exosome clearance efficacy in Figure 5c (2h and 4h). Author should explain for detail and incorporate in the manuscript.

Response: Thank you for your pointing. We added the following sentences to the middle of page 12, as you required, to explain the equal amount of blood exosome clearance efficacy by MSN-AP and MSN-AP- shown in Figure 5c: “However, there was no significant difference between MSN-AP- and MSN-AP in capturing blood A-Exo in vivo (Fig. 5c). The insignificance may be caused by the strong blood shear stress and the in vivo complexity that together interfere the difference between MSN-AP- or MSN-AP in capturing A-Exo in vivo. Interestingly, when a large number of MSN distributed and concentrated in the liver (Figure 5d, 5e far right), where the shear stress is low, and nanomaterial velocity reduces 1000-fold^{25,30}, leading to many times of collisions, interaction and binding between MSN-AP and A-Exo or H-Exo, a significant difference in binding between A-Exo and MSN-AP in the liver was observed (Figure 5e far right).”

15. In the Figure caption of Figure S10, “Trafficking of the positively-charged MSN-Exo- in and out of the same cholangiocyte.” Does it means “negative –charged”?

Response: It should be “positively-charged MSN-Exo”. We changed “MSN-Exo-“to “MSN-Exo”. It is our typo, and we are sorry for the typo.

16. Page 9, line 11 described “The confocal microscopic analysis (Figure 4c) showed the merged yellow dots from the MSN-Exo excreted from the LO2 as an entity.” The yellow dots in figure 4c still inside of cell. It should be a typo. The authors should check the description and the figures throughout the manuscript.

Response: This question is also related to the above 6. Figure 4(c) demonstrates that the endocytosed MSN-Exo by LO2 can be excreted by LO2 as an entity. To demonstrate the free transcytosis of MSN-Exo through LO2, we incubated MSN-AP with A-Exo to form MSN-Exo. The product was centrifuged at 15,000 g and the pellet was incubated with LO2 cells. LO2 cells were centrifuged (at a low rate of 200 g) and collected the precipitate that was the LO2 cells containing MSN-Exo and/or MSN-AP only. We incubated the LO2 cells on the transwell membrane, and collected the cell medium from the lower chamber of the

transwell. The cell medium was added into a confocal dish with LO2 cells on it. Thirty min later, the results were analyzed with a confocal microscope. The confocal microscopic analysis (Figure 4c) showed the merged yellow dots that should be the MSN-Exo excreted from the LO2 as an entity. We double-checked the data and the related presentation, and concluded that it is not a tyro.

We believe that these additions and revisions to the manuscript with our replies to the reviewers' comments in the point-by-point format have largely improved the quality of the manuscript to meet the high publication standards of Nature Communication. We hope that the revised manuscript is now acceptable for publication in this Journal.

Respectfully

Lee Jia, Ph.D.

Editorial Board Member, Current NanoScience

Associate Editor, Current Drug Metabolism

Editorial Board Member, International J. Tumor Therapy

Fellow, American Association of Pharmaceutical Scientists (AAPS)

Reviewers' Comments:

Reviewer #1:

Remarks to the Author:

The authors had answered most of the comments in great detail, revised and expanded the supplemental files and clarified most of concerns that were raised. The overall quality of the manuscript has improved. However, after confirming the newly-added data, several questions came to me.

1. Quantitative relationship in interaction between MSN-AP and A-exo.

In TEM observation shown as Fig.3f, it seems that one A-Exo was surrounded by three MSN-AP. On the other hand, in SEM observation shown as Supplementary Fig.6, schematic images indicate that MSN-AP was surrounded by several A-Exos. Which is correct? Considering that it is impossible to distinguish A-Exo and MSN-AP by SEM observation, I doubt that the schematic image shown in Fig.S6 may be wrong.

In addition, I have a question about the binding capacity of MSN-AP to A-Exo. In Fig.3e, 5 ug of MSN-AP was added to 55 ng or 550 ng of A-Exo, and 5 ug of MSN-AP bound to approximately 3.3 ng and 55 ng of A-Exo. How much of A-Exo can be trapped by MSN-AP?

2. About experimental result shown as Supplementary Fig. 4

I am very surprised to see the result that cationized MSN did not bound to Exosome at all. Considering that zeta potential of cationized MSN and exosomes were approximately +15mV and -30mV, respectively, they should interact each other. The authors should explain this discrepancy.

3. About the unit of y-axis shown in Figs. 5c, 5e and S24.

Do the numbers shown in y-Axis of the Figs that I indicated above mean absolute number of exosomes in blood and in each organ? If so, the numbers of exosome are too small. In most cases, it has been reported that blood concentration of exosomes is more than at least 10^6 .

In addition, x-axis of calibration curve shown as Supplementary Fig23. must be shown as log scale as only 10 fmol of DNA was contained in 1 ug of exosomes. In addition, the relationship between protein amounts of exosomes and number of exosomes must be described.

Reviewer #2:

Remarks to the Author:

The presentation and quality of this revision are improved. I am satisfied with the responses and corrections from the Authors.

Lee Jia, Ph.D. FAAPS.
Director, Cancer Metastasis Alert and Prevention
Center
Distinguished Professor
College of Chemistry, Fuzhou University
Sunlight Building, 6FL, Science Park, Xueyuan
Road, University Town;
Fujian 350116, China
E-mail: pharmlink@gmail.com
cmapcja1234@163.com

September 2, 2019

Re: NCOMMS-19-09320-A

The following are our point-by-point responses **in blue** to the reviewer's comments and suggestions:

Reviewer #1 (Remarks to the Author):

The authors had answered most of the comments in great detail, revised and expanded the supplemental files and clarified most of concerns that were raised. The overall quality of the manuscript has improved. However, after confirming the newly-added data, several questions came to me.

1. Quantitative relationship in interaction between MSN-AP and A-exo.
In TEM observation shown as Fig.3f, it seems that one A-Exo was surrounded by three

MSN-AP. On the other hand, in SEM observation shown as Supplementary Fig.6, schematic images indicate that MSN-AP was surrounded by several A-Exos. Which is correct? Considering that it is impossible to distinguish A-Exo and MSN-AP by SEM observation, I doubt that the schematic image shown in Fig.S6 may be wrong.

In addition, I have a question about the binding capacity of MSN-AP to A-Exo. In Fig.3e, 5 ug of MSN-AP was added to 55 ng or 550 ng of A-Exo, and 5 ug of MSN-AP bound to approximately 3.3 ng and 55 ng of A-Exo. How much of A-Exo can be trapped by MSN-AP?

Response: Thank you for your excellent review. In the previous round of review, your comments and suggestions for revisions are very constructive and helpful. This time, we will try hard to answer your questions and satisfy you.

Both TEM observation (Fig. 3f) and SEM observation (Supplementary Fig. 6) are correct. The reason is that interaction (or binding) between MSN-AP and A-Exo varies depending on their interaction conditions. We prepared MSN-AP and A-Exo uniformly, and used the same batch of MSN-AP and A-Exo to conduct Fig. 3f and Supplementary Fig. 6 experiments for a good reliable comparison. Fig. 3f shows that one A-Exo is surrounded by three MSN-AP in cell medium, while Supplementary Fig. 6 shows that MSN-AP is surrounded by several A-Exo in rat blood. The binding between MSN-AP and A-Exo is random and reciprocal, and happens in a huge volume of medium or blood. Therefore, times of collision between MSN-AP and A-Exo could increase their binding to each other as we demonstrated in Fig. 3e.

Regarding how much of A-Exo can be trapped by MSN-AP: Fig. 3e demonstrated that the formation of MSN-Exo is proportional to the amount of A-Exo (55 and 550 ng) added to the given amount of MSN-AP. We estimate that, in Fig. 3e, there are about 6.6×10^5 and 1.1×10^7 A-Exo trapped on MSN-AP (5 ug) surface in the presence of 55 and 550 ng of A-Exo, respectively.

We sincerely hope that these answers could satisfy you.

2. About experimental result shown as Supplementary Fig. 4

I am very surprised to see the result that cationized MSN did not bound to Exosome at all. Considering that zeta potential of cationized MSN and exosomes were approximately +15mV and -30mV, respectively, they should interact each other. The authors should explain this discrepancy.

Response: You do have reasons to make the hypothesis. However, in their paper entitled "Preferential binding of positive nanoparticles on cell membranes is due to electrostatic interactions: A too simplistic explanation that does not take into account the nanoparticle protein corona" and published in Mater Sci Eng C Mater Biol Appl. 2017; 70 (Pt 1): 889-896, Forest and Pourchez stated that, "It is generally admitted that positive nanoparticles are more uptaken by cells than neutral or negative nanoparticles. It is supposedly due to favorable electrostatic interactions with negatively charged cell membrane. However, this theory seems too simplistic..". They showed that the same nanoparticles in different media possess different zeta potentials, hydrodynamic diameters, and different nanoparticle

protein corona. All these influence their interaction with other biomaterials (see the original table below that we downloaded from this paper for your convenience). This adds a new level of complexity in the interactions with biological systems that cannot be any more limited to electrostatic binding, and other influencing elements need to be considered as well. We proposed that the cationized MSN interact with negative elements in culture media and in blood, which may change MSN potentials, resulting in a change in their electrostatic binding force. We demonstrated that the additional specific intermolecular force between MSN and Exo existed when aptamers on the MSN-AP bound to their corresponding EGFR receptors on A-Exo (Fig. 3j). The supplementary Fig. 4 shows no significant binding between cationized MSN and A-Exo.

Table 1
Physico-chemical characterization of the nanoparticles.

Nanoparticle type	Zeta potential in water (mV)	Zeta potential in DMEMc (mV)	Hydrodynamic diameter in water (nm)	Hydrodynamic diameter in DMEMc (nm)
NP(--)	-30	-96	82 ± 1	104 ± 4
NP(-)	-25	-13	62 ± 5	85 ± 3
NP(0)	0	-11	76 ± 7	88 ± 5
NP(+)	5	-20	75 ± 5	90 ± 5
NP(++)	12	-94	89 ± 2	111 ± 10

We hope that our explanation could satisfy you.

3. About the unit of y-axis shown in Figs. 5c, 5e and S24.

Do the numbers shown in y-Axis of the Figs that I indicated above mean absolute number of exosomes in blood and in each organ? If so, the numbers of exosome are too small. In most cases, it has been reported that blood concentration of exosomes is more than at least 10^6 .

In addition, x-axis of calibration curve shown as Supplementary Fig23. must be shown as log scale as only 10 fmol of DNA was contained in 1 ug of exosomes. In addition, the relationship between protein amounts of exosomes and number of exosomes must be described.

Response: This a very good question! Thank you very much for the question. The blood concentration of exosomes that we estimated is about 10^6 as we explained below: The y-Axis of the Fig 5c, 5e and S24 represents the average number of exosomes identified in 10 randomly-selected single fields of visions.

1) total blood exosome number: we obtained total exosomes 2.2×10^7 from 0.5 mL of blood. The number of 2.2×10^7 is calculated by $(314 \text{ mm}^2 / 0.0576) \times 4000$, in which, 314 mm^2 represents the total observation area of the confocal dish; 0.0576 mm^2 represents a single observation field where we counted about 3000-4000 exosomes (Suppl. Fig. 24).

2) total liver exosome number: we estimated about 1.6×10^7 total exosomes distributed over

the liver. The estimation is made as follows: the total liver volume is 470 mm^3 , a single liver slice is 0.025 mm thick, which give us total slice area 18800 mm^2 ($470/0.025$). Each vision field is 0.0576 mm^2 , resulting in 327000 vision field ($18840/0.0576$), and each field contains about 50 exosomes. Therefore, total liver exosomes are 1.6×10^7 .

3) exosome amounts in other organs can be estimated the same way.

We added the following statement to the legends of related figures “The Y-axis represents the average number of 5-10 randomly–selected single fields of visions”

We have modified the x-axis of **Supplementary Fig. 23** to log scale in our revised version. The relationship between protein amounts of exosomes and number of exosomes was determined by Nanoparticle Tracking Analysis (NTA). The result showed that one μg of exosome protein contains about 2×10^7 exosome particles. We added these words to the revised manuscript page 5, line 10.

We believe that these additions and revisions to the manuscript with our replies to your comments in the point-by-point format have largely improved the quality of the manuscript to meet the high publication standards of Nature Communications. We hope that the revised manuscript is now acceptable for publication in Nature Communications.

Reviewer #2 (Remarks to the Author):

The presentation and quality of this revision are improved. I am satisfied with the responses and corrections from the Authors.

Respectfully

Lee Jia, Ph.D.

Editorial Board Member, Current NanoScience
Associate Editor, Current Drug Metabolism
Editorial Board Member, International J. Tumor Therapy
Fellow, American Association of Pharmaceutical Scientists (AAPS)

Reviewers' Comments:

Reviewer #1:

Remarks to the Author:

The authors adequately responded to my previous comments. However, even after the revision, I have a question about an experimental result shown as Supplementary Fig. 24. In this experiment, tumor-bearing mice was administered with MSN-AP. However, the reviewer cannot understand how blood concentrations of total exosome and tumor exosomes (A-Exo) were measured. If PKH dye was used, the dye stains both A-Exo and exosomes of the other origins, so that blood concentration of A-Exo cannot be determined. Therefore, the method for this experiment must be described in detail. On the other hand, if the concentration of exogenously administered PKH-labeled A-Exo was measured in the data shown as Supplementary Fig. 24, new experiment in which blood concentration of endogenous A-Exo concentration, not the concentration of exogenously-administered A-Exo, as well as blood concentration of total exosomes in tumor-bearing mice are measured must be performed because it is very important to directly prove that MSN-AP can decrease blood concentration of endogenous tumor-derived exosomes under tumor-bearing condition in which tumor-derived exosomes are continuously supplied from the tumor cells. As there is no data that directly prove that MSN-AP can decrease blood concentration of endogenous A-Exo, not the blood concentration of A-Exo administrated by intravenous injection, other than the data shown as Supplementary Fig. 24, this point must be clarified.

Lee Jia, Ph.D. FAAPS.
Director, Cancer Metastasis Alert and
Prevention Center
Distinguished Professor
College of Chemistry, Fuzhou University
Sunlight Building, 6FL, Science Park, Xueyuan
Road, University Town;
Fujian 350116, China
E-mail: cmapcjia1234@163.com

The following is our response (in blue) to the last issue raised by Referee 1

Reviewer #1 (Remarks to the Author):

The authors adequately responded to my previous comments. However, even after the revision, I have a question about an experimental result shown as Supplementary Fig. 24. In this experiment, tumor-bearing mice was administered with MSN-AP. However, the reviewer cannot understand how blood concentrations of total exosome and tumor exosomes (A-Exo) were measured. If PKH dye was used, the dye stains both A-Exo and exosomes of the other origins, so that blood concentration of A-Exo cannot be determined. Therefore, the method for this experiment must be described in detail.

On the other hand, if the concentration of exogenously administered PKH-labeled A-Exo was measured in the data shown as Supplementary Fig. 24, new experiment in which blood concentration of endogenous A-Exo concentration, not the concentration of exogenously-administered A-Exo, as well as blood concentration of total exosomes in tumor-bearing mice are measured must be performed because it is very important to directly prove that MSN-AP can decrease blood concentration of endogenous tumor-derived exosomes under tumor-bearing condition in which tumor-derived exosomes are continuously supplied from the tumor cells. As there is no data that directly prove that MSN-AP can decrease blood concentration of endogenous A-Exo, not the blood concentration of A-Exo administrated by intravenous injection, other than the data shown as Supplementary Fig. 24, this point must be clarified.

Response: Different kinds of cells from different tissues excrete biologically-different exosomes that all exist in the blood. Some exosomes transmit healthy signals, others send unhealthy ones. In the first round of the review process, you requested that “time-course of the amounts of total exosomes and EGFR-positive exosomes in the blood before and after

the single administration of MSNs must be investigated in tumor-bearing mice”, and we did the experiment. We provided you with the added Supplementary Fig. 24, which shows the time-course of the blood total exosomes (including the endogenous EGFR-positive exosomes A-Exo and many other exosomes) and exogenous PKH67-pre-labelled A-Exo. The blood total exosomes without pre-labelling were determined by using the exosome extract kit (Umibio Science and Technology; the kit extracted total exosomes from serum and then stained the exosomes followed by confocal microscopic quantitation), while the exogenous PKH67-pre-labelled A-Exo were directly determined by using confocal microscopy. Since the diluted volume of each sample is known, we can back-calculate the blood total exosomes and the exogenous PKH67-pre-labelled A-Exo for quantification.

You are right in that “the dye stains both A-Exo and exosomes of the other origins, so that blood concentration of (endogenous) A-Exo cannot be determined” by using the dye staining technology. We used the dye staining technology only for determining the “time-course of the amounts of total exosomes and EGFR-positive exosomes in the blood before and after the single administration of MSNs must be investigated in tumor-bearing mice”. The EGFR-positive exosomes were exogenous, and PKH67-pre-labelled.

Nonetheless, we demonstrated that MSN-AP can capture/ decrease the target EGFR exosomes (endogenous and/or exogenous): 1) we used the flow cytometry method to directly quantify endogenous EGFR-positive exosomes in lung cancer patients after the endogenous exosomes were captured by CY5-labelled MSN-AP conjugated onto the beads for flow cytometry analysis (Fig. 6f-g). The data shows that the endogenous EGFR-positive exosomes are very few in the patient blood, and the detection technology is very challenging. Therefore, it is difficult to quantitatively distinguish a few endogenous EGFR-positive exosomes from the “huge” endogenous total exosomes, and hence to quantitatively show how much a few endogenous EGFR-positive exosomes are reduced from the total exosomes after MSN-AP administration. 2) Intravenous administration of MSN-AP (5 mg/kg) every 3 days for 7 times after subcutaneous-implantation of lung cancer cells A549 significantly reduced lung metastases in nude mice in comparison with saline and MSN groups (Fig. 6c).

We appreciate your interests in our work and your careful review of this manuscript. We wish that the above answers to your questions could again satisfy you, and the revised manuscript is now acceptable for publication in Nature Communications.

Respectfully

Lee Jia, Ph.D.

Associate Editor, Current Drug Metabolism
Editorial Board Member, International J. Tumor Therapy

Fellow, American Association of Pharmaceutical Scientists (AAPS)

<https://orcid.org/0000-0001-6839-5545>